



# Oligocene-early Miocene paradox of $p$CO₂ inferred from alkenone carbon isotopic fractionation and sea surface temperature trends

**José Guitián[1*], Samuel Phelps[2, 3], Reto S. Wijker[1], Pratigya J Polissar[2, 4], Laura Arnold[1], and Heather M. Stoll[1]**

[1]Geological Institute, ETH Zurich, Switzerland

[2]Lamont-Doherty Earth Observatory, Columbia University, USA

[3]Harvard University, USA

[4]Ocean Sciences Department, University of California Santa Cruz, USA

*\* present address:* Centro de Investigación Mariña, Universidade de Vigo, GEOMA, Vigo, 36310, Spain

Correspondence to: Jose Guitian (jose.guitian@uvigo.gal; heather.stoll@eaps.ethz.ch)

**Abstract.** Atmospheric carbon dioxide decline is hypothesized to drive the progressive cooling over the Cenozoic. However, at multimillion-year timescales during the Oligocene to Miocene time interval the existing reconstructions, most based on the phytoplankton carbon isotopic fractionation ($\varepsilon_p$) proxy, differ from what is expected to drive the climate observations. Here, we produce two new long-term records of $\varepsilon_p$ over the Oligocene to early Miocene time interval from widely separated locations at IODP Site 1406 and ODP 1168 and increase the resolution of determinations at the equatorial Atlantic ODP 925. These new results confirm a global footprint of $\varepsilon_p$ shift occurring during this interval. Abrupt 3 ‰ declines are found from 27 to 24.5 Ma and 24 to 22.5 Ma, and minimum $\varepsilon_p$ is attained at 19 Ma. Between 28.7 and 29.7 Ma at IODP 1406, a higher resolution sampling reveals orbital scale 100 kyr cyclicity in $\varepsilon_p$. Making use of alkenone-based sea surface temperature (SST) estimates and benthic $\delta^{18}$O estimated extracted from the same samples, we perform a direct comparison with $\varepsilon_p$ to evaluate the relationship with climate dynamics. We observe that across the long Oligocene to early Miocene interval the two sites' relationships contrast with what is expected if CO$_2$ was the main driver of $\varepsilon_p$ and average earth surface temperature evolution was registered at the local surface SST and global benthic $\delta^{18}$O. Moreover, at orbital timescale, $\varepsilon_p$ and benthic $\delta^{18}$O appear to follow an inverse relationship, although located within the multimillion-year period with the strongest direct correlation between these variables (>25.5 Ma). To evaluate the physiological, non- CO$_2$ influences on $\varepsilon_p$, we use modern cultures to evaluate the impact of changing cell size and growth rate on the trends in $\varepsilon_p$. Although at specific time intervals, those drivers seem to explain part of the $\varepsilon_p$ divergence with SST or benthic $\delta^{18}$O, most periods remain largely divergent, particularly the late Oligocene warming. We infer that a common CO$_2$ forcing is likely the dominant control on the coherent temporal trends in $\varepsilon_p$ at widely separated sites, which experienced contrasting temperature evolution and likely experienced different variations in nutrient availability. While CO$_2$ changes likely caused significant changes in radiative forcing, SST variation at the examined sites may have been conditioned by regional heat transport, and the relationship between benthic $\delta^{18}$O and $\varepsilon_p$ could reflect variable phasing between ice growth and global temperature.

**Short summary.** We reconstructed from sediments of different ocean sites phytoplankton carbon isotopic fractionation ($\varepsilon_p$), mainly linked to CO$_2$ variations, during the Oligocene to early Miocene. Records confirm long-term trends but show contrasting relationships with the sea surface temperatures evolution. We evaluate the role of non-CO$_2$ physiological factors such as temperature and nutrients at each site $\varepsilon_p$, highlighting the complexity of interpreting climate dynamics and CO2 reconstructions.



## 1 Introduction

Geological records provide key context to current assessments of the consequences of rising atmospheric $CO_2$ on ice sheet
stability and oceanic temperatures (Foster et al., 2017; Golledge, 2020; Zachos et al., 2008). The Oligocene to Miocene time
interval has been proposed to represent a nonlinear transition between the 'greenhouse' and 'icehouse' stages of Earth history
(Miller et al., 1991; Zachos et al., 2001) useful to evaluate the Earth system climate sensitivity to the hypothesized progressive
$CO_2$ drawdown of the Cenozoic (Deconto et al., 2008; Zhang et al., 2013). However, the long term decline in $CO_2$ estimated
by existing proxy records contrasts with the rather stable climatic state with multimillion year warming and cooling trends
defined by the temperature trends in the deep ocean (Cramer et al., 2011; Lear et al., 2000), and surface ocean (Guitián et al.,
2019; Liu et al., 2009; O'Brien et al., 2020) and with estimated Antarctic Ice sheet volume and sea level (Lear et al., 2004;
Liebrand et al., 2017; Miller et al., 2020).

Most of the Oligocene and early Miocene $p$CO₂ estimates are derived from the sensitivity of marine algae to $CO_2$ (Henderiks
and Pagani, 2008; Pagani et al., 1999, 2000; Pagani et al., 2005; Super et al., 2018; Zhang et al., 2013) based on the carbon
isotopic fractionation in organic matter during photosynthesis ($\varepsilon_p$) of marine phytoplankton (Rau et al., 1996), typically
measured in biomarkers from haptophyte algae. This fractionation can be reconstructed in the past from sediments by the
analysis of $\delta^{13}C$ of the organic lipids and reconstruction of the $\delta^{13}C$ of the DIC in the seawater from which biomass was
produced. Fractionation ($\varepsilon_p$) is predicted to be higher when $CO_2$ availability is high relative to cellular carbon demand. A
decrease in atmospheric $CO_2$ and consequently in $CO_2$ of the surface ocean should therefore lead to a decrease in $\varepsilon_p$ globally.
However, in addition to $CO_2$, the $\varepsilon_p$ in phytoplankton is affected by physiological factors such as the rate of carbon fixation,
which may vary over time in a given location due to variations in temperature or the supply of light. One approach to evaluate
the relative contribution of physiological factors vs $CO_2$ is to produce $\varepsilon_p$ records from sites of widely contrasting oceanographic
setting, where the $CO_2$ signal may be expected to be common to both locations but the environmental factors such as nutrient
availability might not be expected to change in unison. The existing $\varepsilon_p$-based $CO_2$ estimations for the Oligocene are derived
from ~1 m.y. resolution measurements from two sites on the south American margin of the equatorial and South Atlantic; in
the early Miocene an additional North Atlantic record provides data (CenCO2PIP Consortium, 2023). In this study, we produce
a new long-term record of $\varepsilon_p$ over the Oligocene to Miocene time interval at two new, widely separated locations: IODP Site
1406 in the subtropical North Atlantic off the Newfoundland coast, and ODP 1168 in the Southern Ocean off of Tasmania.
We also increase the resolution of determinations at the equatorial Atlantic ODP 925.
65 Our new <1 m.y. resolution $\varepsilon_p$ records from these two mid- latitude locations allow us to directly compare $\varepsilon_p$ with estimates of
SST derived from alkenones extracted from the same samples, since unlike very warm tropical locations, the $U_{37}^{k'}$ index still
retains sensitivity to temperature in the mid-latitudes during the Oligocene and early Miocene. Additionally, we compare $\varepsilon_p$
with benthic $\delta^{18}O$ available from the same sediments, an indicator of high-latitude temperature and Antarctic ice sheet extent
and/or volume. These long-term relationships are contrasted with higher resolution analysis during the middle Oligocene at
70 IODP 1406. The dataset allows a robust evaluation of the relationship between $\varepsilon_p$ and climate dynamics for this time interval.
We further discuss the significance of the observed $\varepsilon_p$ record with the implications for the phytoplankton sensitivity over
multimillion year timescales over the Cenozoic.

### 1.2 An overview of alkenone $\varepsilon_p$ $p$CO₂ proxy

The carbon isotopic fractionation in phytoplankton during photosynthesis is affected not only by the $CO_{2[aq]}$ but also by
75 physiological factors related to the cellular uptake of carbon. These were initially modelled from the assumption of diffusive
carbon acquisition in phytoplankton cells (Rau et al., 1996), where higher $\varepsilon_p$ could be induced by higher $CO_{2(aq)}$, lower
instantaneous growth rates, or a higher cellular surface area to volume ratio. Both cellular permeability and the carbon isotopic
fractionation by the Rubisco enzyme have been assumed to be constant, with Rubisco fractionation typically estimated between
25 and 29 ‰ for alkenone producers (Pagani et al., 2014). Traditional attempts to reconstruct $p$CO₂ from $\varepsilon_p$ have simplified





this original diffusive model by relating $\varepsilon_p$ and $CO_2$ with a single factor $b$ defined to include all physiological parameters affecting the fractionation, and $\varepsilon_f$ representing the fractionation of the Rubisco enzyme (Jasper and Hayes, 1994).

    The $b$-value has been estimated from modern photic zone and culture samples, for which $CO_{2(aq)}$ is independently known. For sedimentary alkenones, previous $pCO_2$ calculations have either, (1) assumed the modern $b$-value for that oceanographic setting remained constant in the past (e.g. Zhang et al., 2013), (2) applied modern relationships between $b$ and phosphate and a

simulated paleo-surface ocean phosphate concentration at the site (Pagani et al., 2011), or (3) estimated the difference between the modern $b$ value at the site and the paleo-setting $b$ value from productivity proxies or proxies for coccolithophore size (Bolton et al., 2016; Henderiks and Pagani, 2007). Despite the appeal of this approach, a recent re-evaluation of cultures and field observations suggest the $b$ term is not well predicted by growth rate, light or cell size alone in a diffusive model. Additional effects occur from carbon concentration mechanisms (CCM) on carbon uptake at lower $CO_2$ concentrations, which

cause a deviation in the $CO_2$ dependence from the theoretical hyperbolic relationship (Hernández-Almeida et al., 2020; Stoll et al., 2019). A further challenge to the physical diffusive model is that the Rubisco fractionation in coccolithophores has been measured in-vitro as 11 ‰ rather than 25‰ (Boller et al., 2011), suggesting that fractionations larger than 11‰ may reflect the operation of additional enzymatic fractionations (Wilkes et al., 2018). The lower Rubisco fractionation has implications for the sensitivity of $\varepsilon_p$ to $CO_2$ (e.g. as explored in (González-Lanchas et al., 2021)).

A meta-analysis of experimental culture data (Stoll et al., 2019) suggests that $\varepsilon_p$ features a logarithmic dependence on $CO_2$, rather than the hyperbolic dependence implied by (Rau et al., 1997). This approach does not resolve the mechanisms for the observed slope of $\varepsilon_p$ dependence on $CO_2$, but over the range of $CO_{2(aq)}$ from 5 to 30 µM, it provides an empirical relationship for interpreting the magnitude of $CO_{2(aq)}$ change implied by a given $\varepsilon_p$ change. The culture dataset illustrates more broadly how $\varepsilon_p$ is the sum of its dependencies on $\ln(CO_2)$, $\ln(light)$, and growth rate and cell radius:


$$(1) \quad \varepsilon_p = 2.66 \ln(CO_2) + 2.33 \ln(light) - 6.98 \, \mu_i - 1.28 \, radius + 6.26$$

where $CO_{2(aq)}$ is in µM, light is in µE, growth rate is day$^{-1}$ and radius is in microns (see Stoll et al. (2019) for confidence intervals on the regression).

From this empirical culture calibration, two challenges remain for the estimation of past $CO_2$ from $\varepsilon_p$ measurements derived from sedimentary alkenones. First, its use would require an estimation of the cell radius, light during the season and depth of alkenone production, and the growth rate. While cell size can be estimated from coccolith length, determining the absolute light and growth rate is rarely possible. Since the equation is a linear sum of these influences, these non-$CO_2$ variables may be integrated into the intercept (e.g. as in González-Lanchas et al. (2021)), where the intercept ($I$) would decrease with higher

growth rates and larger cell sizes and increase with higher light.

$$(2) \quad \varepsilon_p = 2.66 \ln(CO_2) + I$$

    Yet, as with Eq. (1), there remains the challenge of determining which value should be used for the intercept for past conditions,

and whether a constant or variable $I$ is more appropriate for a given site since there are limited proxies for algal growth rate. Recent culture studies document a 0.5 ‰ decrease in $\varepsilon_p$ per 1°C warming (Torres Romero et al., 2024), a magnitude which is indistinguishable from the prediction of growth rate effect on $\varepsilon_p$ and the modeled temperature dependence of coccolithophore growth rates (Krumhardt et al., 2017). This suggests that growth rate, and $I$, may vary over time at a given location if temperature is variable. Therefore, records of SST from alkenone unsaturation or other proxies provide the opportunity to

deconvolve the effects of temperature-driven growth rate variations on $\varepsilon_p$ even when the absolute growth rate is not known.

    In this study, given the potential for oceanographic conditions at the studied sites to differ significantly from those in the modern ocean at these locations, and the concomitant high uncertainties in estimating an appropriate $b$ value for the traditional



approach or *I* for Eq. (2), we do not calculate the past absolute $CO_2$ concentration from our $\varepsilon_p$ measurements. Instead, we account for the potential influence of temperature-driven growth rate changes on our $\varepsilon_p$ records using alkenone temperature

estimates derived from the same samples. Similarly, we evaluate the potential impact of cell size variations on the $\varepsilon_p$ changes. Then, we employ the sensitivity of $\varepsilon_p$ to $CO_2$ in Eq. (2) to estimate possible relative changes in $CO_2$ in the case where the other nutrient-stimulated growth rate or light influences on $\varepsilon_p$ were constant during the studied interval at each site evaluating evidences for this assumption.

## 2 Sites and sediments

We have selected two widely separated paleo locations for this study, from the mid latitude North Atlantic ocean and the high to mid latitudes of the Southern Ocean, from IODP 1406 (40°21.0′N, 51°39.0′W; 3,814 mbsl), ODP 1168 (43° 36.5'S, 144° 24.7'E, and 2463 mbsl) (Fig. 1). A total of 43 and 34 sediment sampling spreading from 30 Ma to 17 Ma at each site were selected. Additionally, higher resolution sampling at IODP 1406 was performed within the 29-30 Ma time window. We also measured an additional six sample set from equatorial Site ODP 925 (4∘12.25'N, 43∘29.33'W, 3042 mbsl) in order to increase

the million-year scale resolution of the previous longest Oligocene record in alkenone carbon fractionation (Zhang et al., 2013). The age model for Sites 1406 and 1168 has been updated using new $^{87}Sr/^{86}Sr$ isotope stratigraphy (Stoll et al., 2024) and the age-modelling software Bacon (Blaauw and Christen, 2011). The new age model for IODP Site 1406 is comparable to previously published chronologies (e.g. as in Guitián et al. (2019) and Van Peer et al. (2017)) but clarifies the duration of the upper Miocene hiatus between 33.3 m and 34.7 m core depths (CCSF-A) as extending from 18.5 to 21 Ma. The ODP Site 1168

age model was revised with the Sr isotope stratigraphy in the interval from 562 to 278 mbsf. The Site 1168 chronology is significantly shifted for most of the Oligocene to early Miocene compared with previous biostratigraphically-based age models and shipboard magnetostratigraphy (Pfuhl and McCave, 2003). Strontium isotope stratigraphy identifies a condensed interval from 22.5 to 21.6 Ma, but suggests sustained sedimentation thereafter through 16 Ma. The age model is most uncertain between 27 and 25 Ma where the Sr isotopic curve has a low rate of change. For the two ODP 1168 samples deeper than the Sr isotope

measurements, and those from Site ODP 925 we use previous age constraints as published previously by Guitián et al. (2020). The paleolatitude reconstruction for the Oligocene to early Miocene barely changes the position of Atlantic sites, in contrast, ODP 1168 moved from 55°S to 48°S between 30 and 15 Ma (Torsvik et al., 2012; van Hinsbergen et al., 2015). Paleodepth estimates for coastal site ODP 1168 suggest a gradual deepening from the Eocene onwards (Exon et al., 2001).

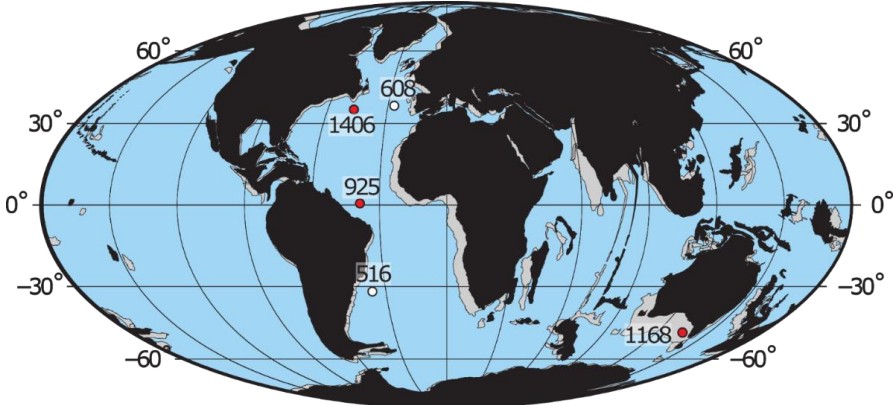


**Figure 1. Location of the study sites. Reconstructed map of continental distribution over the 30 Ma (grey) – 17 Ma (black) time interval. Modified after the plate tectonic reconstruction service ODSN.**



## 3 Methods

### 3.1 Alkenone purification, quantification and δ¹³C analysis

Biomarkers from sediments of IODP 1406 and ODP 1168 were extracted from 30g of freeze-dried sediment using an
Accelerated Solvent Extractor 350 with $CH_2Cl_2$/MeOH (9:1 v/v) solvent for four static cycles at 100°C and further silica gel
column chromatography protocols for purification of the ketone fraction containing the alkenones (see Guitián et al. (2019)
for details).

**Alkenone** ratios were obtained with a Thermo Scientific Trace 1310 Gas Chromatograph (GC)-FID. Originally, for IODP
1406 and 1168 samples the GC-FID was equipped with a non-polar (60 m × 0.25 mm × 0.25 μm) capillary column (ZB-1ms,
Zebron™) at ETH Zurich and at Lamont-Doherty Earth Observatory from Columbia University (Guitián et al., 2019; Guitián
and Stoll, 2021). However, ODP Site 1168 samples older than 22.4 Ma featured more complex chromatograms and a high
diversity of compounds. To reduce the effects of coelution, samples were additionally analyzed on a 105m column RTX-
200ms at ETH Zurich, which improved separation of long chain ketones (Rama-Corredor et al., 2018). The following
temperature program was used: 1 min at 50°C, temperature gradient of 40°C/min to 200°C and 5°C/min to 300°C, hold for
45min, and increased to 320°C at 10°C/min and hold for 8min. Carrier gas was Helium at a flow rate of 1.5ml/min. In-house
standards and replicates injected at every sequence ensured instrument precision. A subset of IODP Site 1406 and samples
younger than 22.4 Ma from Site 1168 were re-measured with the RTX-200 to ensure replicability (Table S1). Method used for
each organic analysis is described in the supplementary material dataset.

**Sea surface temperature** was calculated from $U^{k\prime}_{37}$ ratio using the Bayspline calibration (Tierney and Tingley, 2018) for all
samples in IODP 1406 and for the young set of samples in ODP 1168. Because for those, we find $U^{k\prime}_{37}$ ratios within the
analytical uncertainty using both columns, we report the original ZB-1 results for all Site 1406 samples and Site 1168 samples
younger than 22.4 Ma / 408.22 m depth. The RTX-200 column provided substantially improved resolution of C38 peaks,
allowing quantification of $C_{38:2}$ and $C_{38:3}$ ME peaks, but for samples between the ages of 23.1 and 29.1 Ma in ODP 1168 it did
not perform well enough for the C37 peaks. Therefore, for this set we provide temperatures estimated from the $U^{k\prime}_{38ME}$ ratio
applying the Novak et al. (2022) core top calibration. For the ODP 925 equatorial site samples, the $C_{37:3}$ methyl ketone is under
the detection limit, therefore we further purified and analyzed the extracted organics as in Guitián et al. (2019) to get the
temperatures from the TEX86 ratio using the BAYSPAR calibration by (Tierney and Tingley, 2015).

**Compound-specific δ¹³C** measurements were performed on a Thermo Scientific Trace 1310 Gas Chromatograph coupled to
a Thermo Scientific GC Isolink II, a Conflo IV, and a Delta V Plus Mass Spectrometer at ETH Zurich. Oxygen was flushed
through the combustion reactor for one hour at the beginning of each sequence and seed oxidized for one minute before each
injection. Alkenones from ODP Site 1168 and IODP 1406 were analyzed on a GC equipped with a non-polar capillary column
(60 m × 0.25 mm × 0.25 μm) (ZB-1ms, Zebron™) and 5-m guard column. Helium was used as carrier gas flow with 2-ml/min.
GC oven was set at 90°C ramped to 250 at 25°C/min, to 313°C at 1°C/min and finally 320°C at 10°C /min. The GC oven
was then maintained isothermally for 20 min. A subset of IODP 1406 samples were measured additionally at the Lamont-
Doherty Earth Observatory on equivalent instrumentation but with some modifications for improved sensitivity (Baczynski et
al., 2018) and following similar GC procedures, with similar results.

From ODP Site 1168, a subset of samples older than 24.5 Ma featuring more complex chromatograms were rerun on a GC-
irMS equipped with a RTX-200 ms column. GC oven was set at 50°C ramped to 275°C at 40°C/min, to 295°C at 0.5°C, hold
for 22 min, and finally ramped to 320°C at 10°C/min and hold for 5 min. Flow rate was 1.5ml/min. Comparison of a subset of
samples from Site 1406 and ODP 1168 younger than 22.4 ma showed that δ¹³C C37:2 were similar on both ZB-1 and RTX-
columns. We consequently report here δ¹³C C37:2 from the ZB-1 runs, with the exception of the samples from Site 1168
older than 22.4 Ma. All values are reported here in parts per mil (‰) relative to VPDB (Vienna Pee Dee Belemnite). Sample





replicates, in-house alkenone standard (provided by G. O'Neil, Western Washington University, and C. M. Reddy, Woods
Hole Oceanographic Institution), and known isotopic mixtures A5 and B4 (supplied by A. Schimmelmann, Univ. of Indiana)
were simultaneously measured to determine the analytical accuracy of the measurement and an uncertainty of 0.5 ‰.

### 3.2 Estimation of aqueous carbon dioxide δ¹³C

Isotopic composition of $CO_{2[aq]}$ is estimated from the temperature dependent fractionation between DIC and aqueous $CO_2$
during alkenone production of Rau et al. (1996) based on Mook et al. (1974) and Freeman and Hayes (1992):

(3) $\delta^{13}C_{[CO2]aq} = \delta^{13}C_{DIC} + 23.644 - \left(\frac{9701.5}{T}\right)$

The δ¹³C DIC may be estimated from the δ¹³C of calcium carbonate of benthic foraminifera with the assumption of a constant
and known offset between the δ¹³C DIC of the deep and surface ocean, or from bulk sediment calcium carbonate that is mostly
derived from coccolithophores and planktonic foraminifera. Site 1406 and 925 features sufficient well preserved benthic
foraminifera, mainly epifaunal *Cibicidoides* spp. larger than 200 µm, δ¹³C of surface ocean DIC can be calculated here applying
a constant offset of 2 ‰ of measurements from the same samples following previous Miocene and Oligocene studies (Guitián et al.,
2019; Pagani et al., 2011; Zhang et al., 2013). However, at ODP Site 1168 benthic foraminifera were scarce for picking
for isotopes in many intervals and the progressive evolution of water depth at the site may change the δ¹³C offset between the
benthic environment and the surface ocean over time. Consequently, to follow the same approach for all studied records we
calculate the δ¹³C DIC from the δ¹³C of the bulk carbonate, which is dominated by *Reticulofenestra* coccoliths (Guitián et al.,
2020). Because there is no divergence of vital effects between small and large coccoliths in the late Oligocene to early Miocene
(Bolton and Stoll, 2013), we propose that the offset between coccolith δ¹³C and DIC is likely to remain constant. We subtract
0.5 ‰ from the δ¹³C$_{bulk}$ to calculate δ¹³C$_{DIC}$, based on average alkenone-producing coccoliths cultured at DIC <4 mM compiled
in Stoll et al. (2019). Support for estimating photosynthetic fractionation from coccolith δ¹³C is provided by recent culture
studies of *G. oceanica* (Torres Romero et al., 2024).

Stable isotopes of carbonates were measured as described in in Guitián et al. (2019) with the guidelines from Breitenbach and
Bernasconi (2011) for small carbonate samples on a GAS BENCH II Delta V Plus irMS from Thermo Scientific with
international (NBS-19 & 18) and in-house carbonate as standards achieving a precision of 0.07 ‰.

### 3.3 Calculation of $\varepsilon_{p\ 37.2}$

Carbon isotopic fractionation ($\varepsilon_p$), describes the fractionation occurring during photosynthesis when carbon is fixed into algal
cellular biomass (δ¹³C$_{org}$) from the ambient aqueous $CO_2$ (δ¹³C$_{CO2aq}$) (Freeman and Hayes, 1992):

(4) $\varepsilon_p = \left(\frac{(\delta^{13}C_{[CO2]aq} + 1000)}{(\delta^{13}C_{org} + 1000)} - 1\right) * 1000$

Organic δ¹³C is obtained from the δ¹³C analysis of haptophyte specific alkenone di-unsaturated C$_{37.2}$. Culture experiments
showed that the lipid organic matter is depleted in ¹³C relative to the whole cell isotopic composition by 4.2 ‰, a correction
that needs to be applied (Popp et al., 1998; Wilkes et al., 2018):

(5) $\delta^{13}C_{org} = [(\delta^{13}C_{37:2} + 1000) * ((4.2/1000) + 1) - 1000]$

Uncertainties were propagated by a full Monte Carlo (n = 10000) simulation following (Tanner et al., 2020).

To compare our new records with previous data spanning the same time interval, we discuss published $\varepsilon_p$ datasets recently
compiled by the paleo $CO_2$ community (CenCO2PIP Consortium, 2023), from DSDP 516 in the South Atlantic (Pagani et al.,
2000; Pagani et al., 2011; Pagani et al., 2005), DSDP 608 in the North Atlantic (Super et al., 2018), and the equatorial site
from ODP 925 (Zhang et al., 2013). For these, we ensure that $\varepsilon_p$ for the published records is calculated from biomarker-based



paleothermometers. The most recent publications from DSDP 608 and 925 used GDGT-derived estimations from TEX-86. To better compare our results with DSDP 516, where originally temperatures were derived from $\delta^{18}O$ of planktic foraminifera for the Miocene section and GDGTS for part of the Oligocene, we have updated the $\varepsilon_p$ calculations using a running averaging of the recent higher resolution GDGT temperature reconstructions from Auderset et al. (2022) at the same site.

For our data of paired $\varepsilon_p$ and alkenone SST, we calculate the shift in $\varepsilon_p$ which is expected from temperature-stimulated growth rates. We adjusted each samples $\varepsilon_p$ absolute value by using the relationship of -0.48 ‰ per 1°C SST difference relative to the record average (Torres Romero et al., 2024). We complete a similar exercise for cell radius, calculating the deviation in $\varepsilon_p$ relative to the median cell size, for each point using the culture dependence of $\varepsilon_p$ on cell radius (Stoll et al., 2019). Biogenic silica (bioSi) was determined on 20 samples from ODP 1168 following methods described previously in Guitián et al. (2020).

**4 Results and Discussion**

**4.1 Trends in $\varepsilon_p$ in the Oligocene to early Miocene**

**4.1.1 New $\varepsilon_p$ records from sites 1406 and 1168**

In both long-term records from Site 1406 and Site 1168, $\delta^{13}C$ of $C_{37.2}$ alkenones range from very low values near -30 ‰ in the early Oligocene (28-30 Ma) increasing to -24 ‰ by 18-20 Ma (Fig. 2). The new calculated $\varepsilon_p$ decrease from the Oligocene to

the early Miocene, defined most precisely at the highest resolution North Atlantic Site 1406, features abrupt 3 ‰ declines from 26.5 to 25.4 Ma and 24 to 22.8 Ma. Newly obtained ODP 925 $\varepsilon_p$ determinations within the interval 25 Ma to 19 Ma are in agreement with previous determinations at this site (Zhang et al., 2013) resolving steps at 26 ma and 21 Ma. The trends in $\varepsilon_p$ calculated from benthic $\delta^{13}C$ are similar to those calculated from the coccolith-dominated bulk $\delta^{13}C$. In the high resolution section from 28.8 to 29.6 Ma in Site 1406 there is no long-term trend, but orbital scale $\varepsilon_p$ variations exceed 1.5 ‰ in amplitude

(Fig. 3, Fig. S1). Several ~100 ky orbital scale variations of 0.75 ‰ benthic $\delta^{18}O$ and bulk $\delta^{18}O$ are sampled, as expected for this time window of high 100 ky power in benthic $\delta^{18}O$ in other sites (Liebrand et al., 2017).

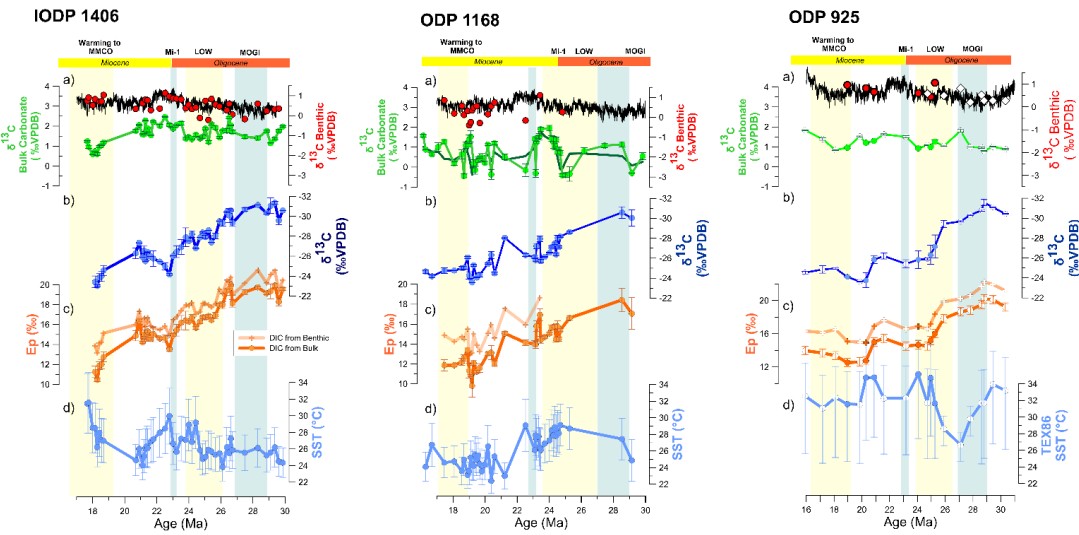

**Figure 2: Analytical results of this study. a) Carbonate stable isotopes for benthic foraminifera (red symbols), data from this work for each site; black line shows results from Westerhold et al., (2020) and bulk sediment carbonate (green color). b) Alkenone $C_{37.2}$**
**$\delta^{13}C$. c) Calculated alkenone carbon fractionation, solid line DIC $\delta^{13}C$ is derived from bulk carbonate, transparent line from picked benthic foraminifera at the same samples. d) Temperature estimates. IODP 1406, including SST and benthic $\delta^{18}O$ dataset from Guitian et al., (2019); ODP 1168, dark green $\delta^{13}C$ bulk carbonate shows the 4-point moving average used to calculate $\varepsilon_p$ at the site; ODP 925, filled circles are new measurements for this study, white symbols are published data (Zhang et al., 2013), being $\varepsilon_p$**



recalculated following method described in text. Note SST is derived from GDGT at this site. SST errors bars: 1406- 1σ, 1168-$U_{37}^{k'}$-
1σ, 1168-$U_{38ME}^{k'}$-2σ, 925-TEX$_{86}$-2σ.

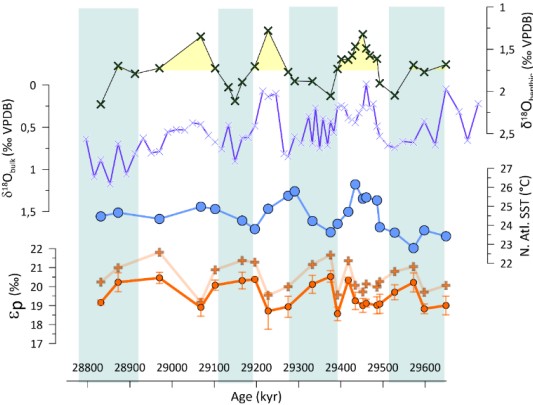

**Figure 3.** High resolution sampling from IODP 1406, showing δ$^{18}$O of bulk carbonate and benthic foraminifera, alkenone SST estimates, and ε$_p$ calculated from bulk carbonate (circles) and benthic foraminifera assuming a constant offset (crosses).

**4.1.2 Comparison of 1168 and 1406 ε$_p$ records with published Atlantic records**

The overall decline in ε$_p$ through the time interval of our records, is broadly comparable to the trend in published ε$_p$ datasets recently compiled by the paleo CO$_2$ community (CenCO2PIP Consortium, 2023), which exhibits a long term decrease in the late Oligocene through the transition to the Miocene and an overall low and stable early Miocene (Fig. 4). However, several factors complicate a detailed comparison of our new and the previously published records.

Rapid ε$_p$ shift comparison might be hindered by site chronology uncertainties. Although all records are presented here on the
GTS 2012 (Gradstein et al., 2012), ODP 1168 and IODP 1406 age models rely on Sr isotope stratigraphy (Stoll et al., 2024), whereas ODP 925, DSDP 516 and ODP 608 are based exclusively on biostratigraphic and magnetostratigraphic reversals datums (Curry et al., 1995). As seen in sites 1168 and 1406, Sr isotopic stratigraphy can adjust age determinations by 0.5 to 1 Myr. or even up to 2 Myr. at few cases.

Additionally, differences in the absolute value of ε$_p$ among records may also reflect contrasting approaches to the
reconstruction of DIC δ$^{13}$C in the different studies. At DSDP 608, the DIC δ$^{13}$C was reconstructed from surface-dwelling foraminifera *G. quadrilobatus* (Pagani et al., 1999), while at DSDP 516 the Miocene section was estimated from planktic foraminifera and most of the Oligocene samples DIC δ$^{13}$C was determined from fine fraction (Pagani et al., 2000; Pagani et al., 2005). Published ODP 925 ε$_p$ has been recalculated here with DIC δ$^{13}$C derived from bulk carbonates of nearby samples, to resolve the previous divergent estimates from planktic and benthic foraminifera (Zhang et al., 2013).

The longest record from **DSDP Site 516** exhibits a general ε$_p$ decline from the Oligocene to early Miocene. However, due to lower resolution at this site, we cannot evaluate if there is an abrupt 3 ‰ decline from 26 to 24.5 Ma as seen in sites 1406 and 925. A steep ε$_p$ decline between at 21 and 20 Ma in Site 516 may be within age uncertainty of the decrease observed at ODP 1168 and ODP 925; additional Sr isotope stratigraphy at Site 516 in this time interval could help test the synchronicity. The late Oligocene at DSDP 516 features a 5‰ peak in ε$_p$ between 24.5 and 24.9 Ma, which is not reflected at 1406, or 1168 sites.

The early Miocene record at DSDP **Site 608** shows a more variable ε$_p$ with a much steeper decline through the early Miocene and higher amplitude variation compared to other sites. The characteristic minimum in ε$_p$ from 18 to 17 Ma is potentially within the age model uncertainty of the minimum in ε$_p$ at 19 Ma in 1168 and the minimum identified at Site 1406 at 18.5 Ma. Updated stratigraphy could contrast more robustly the timing of these events. If there is high amplitude short term variability



in $\varepsilon_p$ in the early Miocene as in the Oligocene time interval (Fig. 2), there is also the potential for low resolution sampling to

undersample high frequency temporal variability and generate aliasing artefacts.

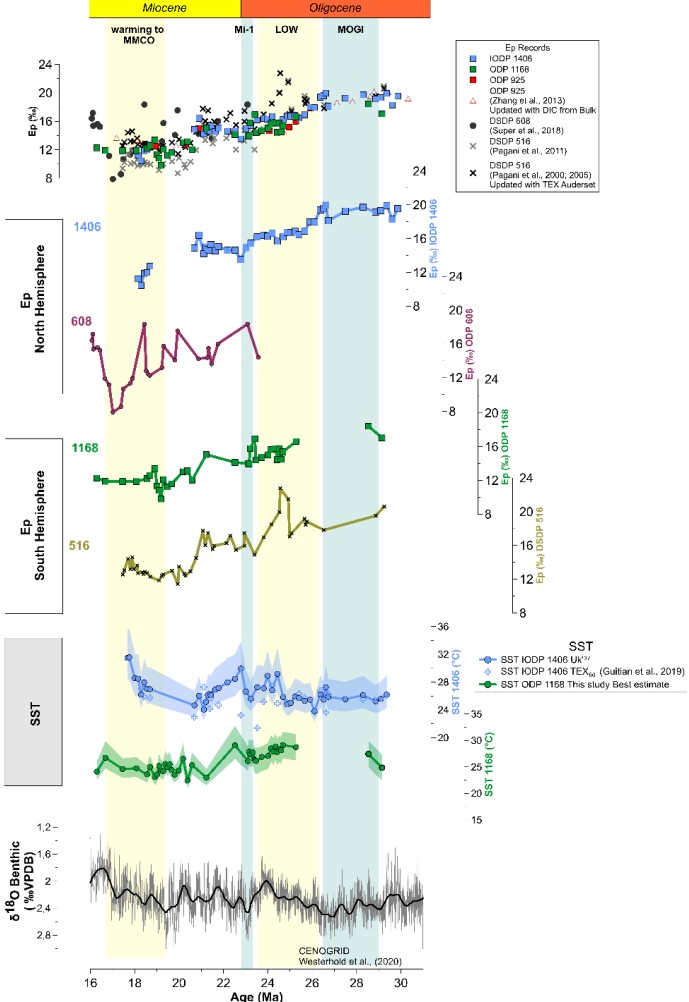

**Figure 4: Oligocene to Miocene global long term $\varepsilon_p$ trends. Comparison of the new obtained $\varepsilon_p$ records with previously published alkenone measurements. All $\varepsilon_p$ estimates have been recalculated following the methodology described in text with data source described on Table S2.**

**4.2 CO₂ vs size and nutrient effects on $\varepsilon_p$**

In addition to $CO_2$, $\varepsilon_p$ may be influenced by changes in cell surface area to volume ratio and cellular growth rate regulated by light and nutrients. There is no long-term trend in mean coccolith size in these records (Guitián et al., 2020) and estimating the impact of it on the $\varepsilon_p$ records shows a negligible effect on long-term trends (Fig. 5). At discrete time intervals of IODP Site 1406, the size effect reduces most values older than 27 Ma and produce a steeper decrease on $\varepsilon_p$ towards the Oligocene Miocene

transition.

For the statistical model of Eq. (1), it is complex to identify proxy records for any possible effect of nutrient-stimulation of growth rate or changes in the mean light conditions at the depth of growth. In modern spatial gradients in the ocean, these factors are often coupled, so that settings characterized by deep mixing and high nutrient supply rates to stimulate growth, are also characterized by lower mean light levels due to the deeper mixing, both factors lowering $\varepsilon_p$.





As one possible nutrient indicator, a higher concentration of biogenic silica (bioSi) in sediments may reflect a higher rate of bioSi delivery to the seafloor due to higher export production produced by siliceous organisms (mainly diatoms) in the ocean (Ragueneau et al., 2000). In the modern ocean, regions with abundant dissolved Si in the photic zone are regions also characterized by higher concentrations of macronutrients such as P and N. At IODP 1406, bioSi concentrations generally increase from the Oligocene to earliest Miocene, potentially indicating a gradual increase in the concentration of dissolved Si

in surface waters at the site (Fig. 5). If the increase in dissolved silica observed at the North Atlantic is correlated to an increase in dissolved P or N, it could contribute to increase in growth rate, and therefore likely increase in biomass and chlorophyll, which would reduce light in the water column both being part of the observed long term decrease of $\varepsilon_p$. However, the actual correlation between bioSi and $\varepsilon_p$ is not that strong (Fig. S2), suggesting that while increased nutrient concentrations could contribute to the long-term evolution of $\varepsilon_p$, the specific steps of $\varepsilon_p$ decline are less likely to be driven by increased nutrients

and growth rate.

The drivers for increasing bioSi burial rates at Site 1406 are not clear, although they could reflect a global increase in nutrient delivery. Important changes in the rate of continental weathering within the Oligocene- early Miocene are often interpreted from the evolution of radiogenic isotopes of Sr, Li and Os (Misra and Froelich, 2012) including the steep rise in $^{87}Sr/^{86}Sr$, although the precise origin of the late Eocene and Miocene increase in $^{87}Sr/^{86}Sr$ remains under discussion (Rugenstein et al.,

2019).  On a global scale, the nutrient delivery may be conditioned by the riverine supply of P from continental erosion and weathering of P containing minerals. On the time scales examined in our records, much longer than the residence time of P, the net effect on nutrient concentrations depends on the balance of the supply and the nutrient removal in sediments.

While a significant increase in erosion and weathering and nutrient inventory is one mechanism to contribute to the long term decline in $\varepsilon_p$ via enhanced algal growth rates, an increase in erosion and weathering can itself contribute to a $CO_2$ drawdown

by $CO_2$ consumption through silicate weathering and enhanced burial of organic carbon in delta regions (Raymo and Ruddiman, 1992). If the biogenic Si increase at 1406 were representative of a global trend, an increase in nutrient supply may have contributed to $\varepsilon_p$ decline through both $CO_2$ decline and increased nutrient stimulation of phytoplankton growth.  A global decline in $\varepsilon_p$ solely from increased weathering and nutrient concentrations without a $CO_2$ decline would require that in the Oligocene, the nutrient release from silicate weathering was less coupled to carbon burial than in the late Neogene. If the

periodically glaciated margin of Antarctica is a major locus for increased erosion and weathering in the Oligocene (Reilly et al., 2002), release of nutrients and radiogenic isotopes may have occurred in the continental margins, but with much less organic carbon burial than the modern Himalaya system due to limited terrestrial biomass on Antarctica and temperature and sea-ice limited oceanic biomass production in the marine regions.





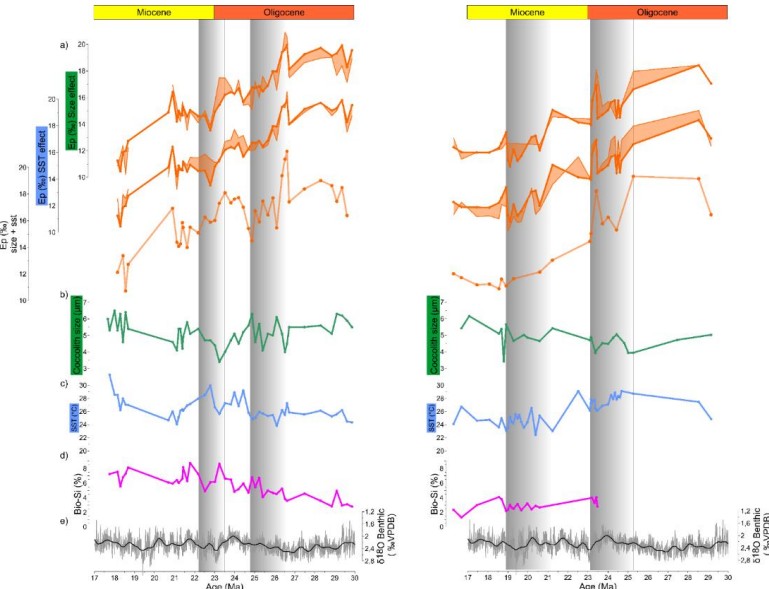

**Figure 5: Timeseries of $\varepsilon_p$ at IODP 1406 and ODP 1168 including a) the measured ep record (solid line) and the estimated $\varepsilon_p$ resulted once size and temperature effect is applied following Torres et al., (2024), and Stoll et al., (2019) (transparent shadows). b) Coccolith size record from Guitian et al., (2020); c) SST estimates d) Biogenic silica measurements (Guitian et al., (2020) and this study); e) reference CENOGRID benthic $\delta^{18}O$ curve (Westerhold et al., 2020)**

On the other hand, the long term trend of increased bioSi is not observed in the Southern ocean Site 1168 (Fig. 5). The available Miocene bioSi at ODP 1168 is stable with no change across the steep $\varepsilon_p$ drop from the latest Oligocene to early Miocene. The increasing distance of Site 1168 from the coastline with basin subsidence may have decreased the availability of Si from the early Oligocene through the early Miocene, imparting a local effect superimposed on any potential global trend. However, likely not only Si but also other nutrients would decrease with increasing distance from the coast. If the long term trend in $\varepsilon_p$ at both 1406 and 1168 sites were conditioned by increased nutrient availability, faster growth rates, and lower light levels it would require bioSi accumulation rates at Site 1168 to be decoupled from the overall changes in nutrient availability, which we consider less likely. Consequently, we propose that the similarity in trend and magnitude of the long term $\varepsilon_p$ decline in both sites (and in tropical Site 925), is more consistent with a global forcing of $\varepsilon_p$, which may be most plausibly driven by a significant decrease in atmospheric $CO_2$ and $CO_2$aq.

### 4.3 Relationship between $\varepsilon_p$ and SST and benthic $\delta^{18}O$

The new $\varepsilon_p$ data from Site 1406 and Site 1168 provide the first records of $\varepsilon_p$ from the early Oligocene to early Miocene with alkenone unsaturation indices as independent estimations of SST for the precise time intervals of $\varepsilon_p$ determination. Since they are biomarkers derived from the same organism, alkenone-derived SST estimates correspond to the same season and growth depth as the alkenone $\varepsilon_p$ determinations. There are two processes which may influence the relationship between temperature and $\varepsilon_p$. First, higher temperatures lead to higher phytoplankton carbon fixation rates, decreasing $\varepsilon_p$. Secondly, higher $CO_2$ would increase $\varepsilon_p$ and through radiative forcing lead to warmer global average temperature and SSTs. The expected relationship between $\varepsilon_p$ and SST from either process could be obscured by a superposition of temperature effects on growth rate and a climatic correlation of $\varepsilon_p$ with mean air temperature.

#### 4.3.1 Million year scale relationships

Across the overall time interval, Site 1406 $\varepsilon_p$ is weakly inversely correlated with SST, whereas Site 1168 $\varepsilon_p$ is weakly positively correlated with SST (Fig. 6; Fig. S3, Table S3). Our estimation of the growth rate effect due to warmer temperatures shows





that it has a very limited impact on the long term $\varepsilon_p$ trend, amplifying slightly the long-term excursion in ODP 1168 and imparting a minor increase in $\varepsilon_p$ in the late Oligocene 25.5 to 24 Ma in IODP 1406 but otherwise not affecting the sign of the overall trend (Fig. 5, 6, Fig. S3). At our studied sites across the 30 to 17 Ma time interval, the long term average warming of 2-3°C is insufficient to account for the 7 ‰ decline in $\varepsilon_p$ due to a temperature-driven growth rate effect.

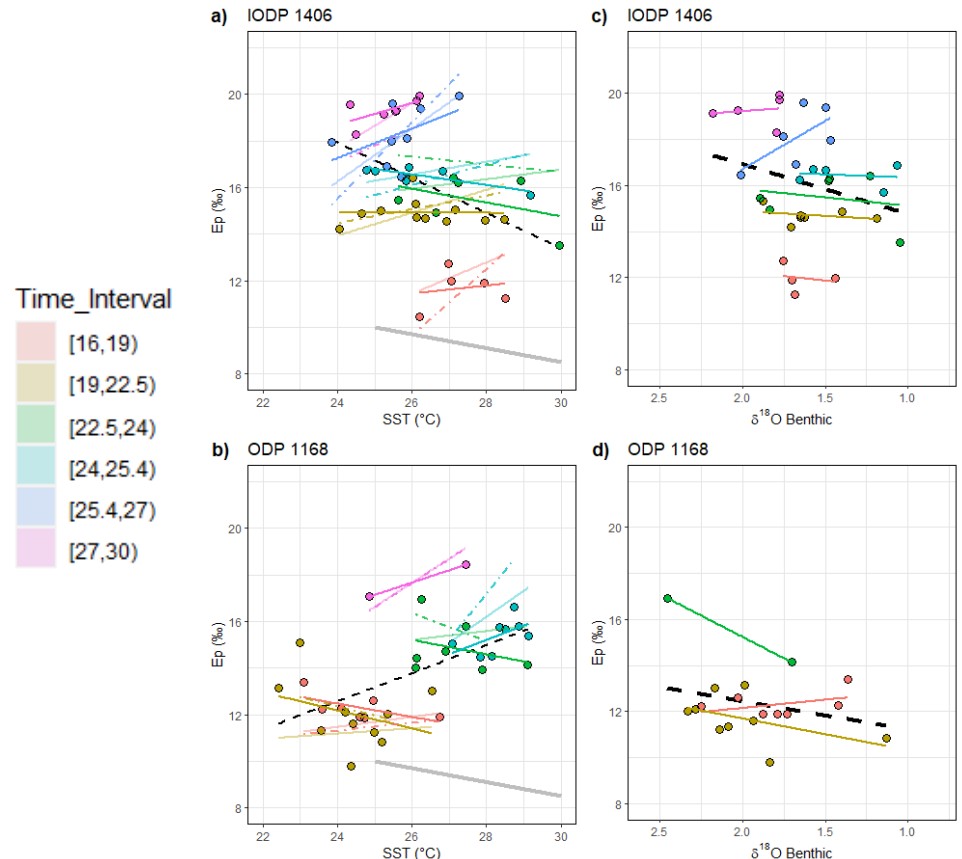


**Figure 6. Relationship between calculated $\varepsilon_p$ and SST and benthic $\delta^{18}$O for the same samples from IODP 1406 and ODP 1168. Dashed dark line in the plot background shows the regression of the overall dataset. Grey line in a) b) shows empirical temperature-growth rate effect in temperature described in the text (Torres et al., 2024) as the $\varepsilon_p$ change with growth rate if $CO_2$ were constant. Colored lines shows the regression at specific time intervals for the measured $\varepsilon_p$ (solid), $\varepsilon_p$ with the temperature effect removed (transparent), and $\varepsilon_p$ with both the temperature and size effect removed (dashed). See Fig. S3 for overall dataset relationships.**


Yet aside from the overall inverse correlation at IODP 1406, during certain time intervals the correlation is positive. Before 25.4 Ma, there is a general positive correlation between $\varepsilon_p$ and alkenone SST (Fig. 6, Table S3); for the intervals 27 to 30 and 25.4 to 27 Ma in Site 1406 with a slope of **0.7** (n=6, n=10). This is also true for the 25.4 to 24 Ma interval in 1168 with 0.6

slope (n=6). This slope is significantly greater when the influence of size and temperature-growth rate effects on $\varepsilon_p$ are removed. In other time intervals, there is a negligible slope or negative slope with measured $\varepsilon_p$, for example -0.3 (24 to 22.5 Ma in both sites, n=7, n=4) or during the drop after 19 Ma at IODP1406 appear to occur at times of warming temperature (slope= 0.2, n=5) that slightly improves when the $\varepsilon_p$ is adjusted to size and temperature.

Over the studied time interval, this relationship similarly shows insignificant correlation for the previously published $\varepsilon_p$,

records with updated age models and $\varepsilon_p$ calculations (Fig. S4), although temperatures estimates are derived from GDGT which might not reflect the same depth and/or season of coccolithophore growth. Negative covariance is observed at DSDP 608 from



19 to 16 ma, at DSDP 516 before 21 Ma and for the few samples from 27 to 24 ma at ODP 925. Some of these intervals feature significant temperature changes of 4 to 5°C, and therefore the temperature-growth rate effect on $\varepsilon_p$ may be significant, and the negative slopes observed in some intervals are consistent with this being the dominant effect (gray line in Fig. 6a and 6b). At

1406, during the older intervals of positive correlation of SST and $\varepsilon_p$, potentially the growth rate stimulation due to higher SST was balanced by a decrease of nutrient availability during warmer temperatures as suggested by the bioSi evolution (Fig. S2), whereas during younger time intervals, temperature exerted a dominant effect on growth rate.

Benthic $\delta^{18}O$ was measured in multiple time intervals in Site 1406. Benthic $\delta^{18}O$ has been proposed to reflect global surface temperature (Evans et al., 2024; Hansen et al., 2013) and as such may be less sensitive than SST to regional reorganizations

of heat transport. Alternatively, benthic $\delta^{18}O$ has been proposed to be highly sensitive to the areal extent of the Antarctic ice sheet due to its cooling effect on surface ocean temperatures in regions of deepwater formation (Bradshaw et al., 2021; Lisiecki and Raymo, 2005; Shackleton, 1987). If the global surface temperature change translated to changes in surface ocean temperatures at Site 1406 and Site 1168, we would expect the temperature-growth rate effect to generate a direct correlation between benthic $\delta^{18}O$ and $\varepsilon_p$. If the radiative forcing effect on global temperature change were dominant, we would expect an

inverse correlation between $\varepsilon_p$ and benthic $\delta^{18}O$. As for SST, only the time intervals older than 25.4 Ma exhibit the inverse correlation expected from radiative forcing, whereas other intervals suggest neutral slope which may reflect the superposition of growth rate and $CO_2$-radiative effects on $\varepsilon_p$.

### 4.3.2 Relationships between $\varepsilon_p$, temperature and benthic $\delta^{18}O$ at orbital timescales

In the high resolution sampling between 29.0 and 29.6 Ma, despite a significant 1 ‰ range in $\delta^{18}O$ benthic and $\delta^{18}O$ bulk, we

likewise observe no inverse relationship between $\varepsilon_p$ and $\delta^{18}O$ benthic, or between $\varepsilon_p$ and $\delta^{18}O$ bulk (Fig. 7). We also observe no significant correlation between $\varepsilon_p$ and alkenone SST. Because the magnitude of SST variation is small over this time interval, the impact of temperature-stimulated carbon fixation rates is not a significant impact on the relationship between $\varepsilon_p$ and any of these variables – a temperature-corrected $\varepsilon_p$ record for the 29 to 29.6 Ma interval would still not exhibit an inverse relationship between $\varepsilon_p$ and $\delta^{18}O$ benthic as observed in the late Pleistocene glacial cycles (Hernández-Almeida et al., 2023).

If $\varepsilon_p$ variations are dominantly responding to $CO_2$, our results suggest that low $CO_2$ is not contributing to greater ice volume and/or colder ocean temperatures on 100 ky cycles and that the relationship between Antarctic ice growth and $CO_2$ may be more complex at this time.

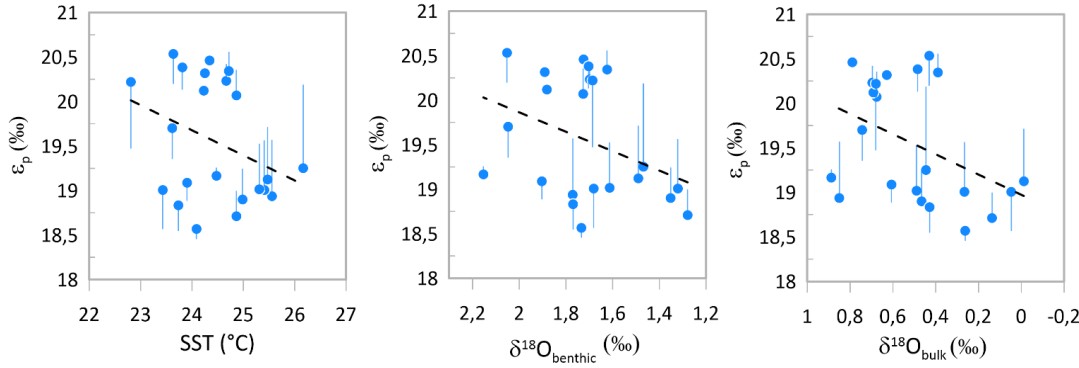

**Figure 7. Relationship between calculated $\varepsilon_p$ and temperature (r2=-0.34), $\delta^{18}O$ from benthic foraminifera (r2=0.42), and $\delta^{18}O$ from**
**bulk carbonates (r2=0.37) for the high resolution interval samples from IODP 1406. Vertical error bars shows the temperature effect**
**on $\varepsilon_p$ following findings from cultures (Torres et al., 2024). Dashed line shows the linear regression for all plotted samples.**



### 4.4 A Climate and $CO_2$ paradox from the Oligocene to early Miocene

The long term trends between 30 and 16 Ma based on new $\varepsilon_p$ data at two sites and recalculation of previous $\varepsilon_p$ studies with uniform methods cannot be attributed to a temperature effect on growth rate and $\varepsilon_p$, nor to changes in the cell size of the alkenone producing community. Both effects are small in magnitude according to the sensitivities observed in cultures and do not alter the long-term trend (Fig. 5). Therefore, the long-term $\varepsilon_p$ decline must have a significant global driver, with the most obvious being a decline in $p$CO$_2$.

Although the calculation of absolute $CO_2$ concentrations from $\varepsilon_p$ in the Oligocene and early Miocene remains challenging, the logarithmic dependence of $\varepsilon_p$ on $CO_{2[aq]}$ observed in cultures allows us to estimate the relative changes in $CO_2$ if the sensitivity of $\varepsilon_p$ to $CO_2$ in the Oligocene were similar to modern cultured species. If we incorporate a temperature correction on growth rate (Krumhardt et al., 2017) equivalent to the magnitude from cultures (Torres Romero et al., 2024) and apply the 50[th] percentile estimate of the modern culture $\varepsilon_p$ dependence on $\ln [CO_{2[aq]}]$ of 2.66, it implies major changes in $CO_2$ concentrations, with potentially 4 halvings of $CO_2$ concentration from 29 to 16 Ma (Fig. 8). Modern General Circulation Models (GCM) estimate climate sensitivity at 3 to 5°C per doubling or halving of $CO_2$, which if representative for the Oligocene to early Miocene, would imply 12 to 20°C of cooling of earth's mean surface temperature. Although ocean is 70% of the globe and temperature changes are 1.3- to 1.8-fold less than land temperature (Sutton et al., 2007), such a large temperature change of at least 10.4°C would be expected to be reflected in paleoceanographic proxies. Application of the lower confidence interval of modern culture $\varepsilon_p$ dependence on $\log(CO_{2[aq]})$ of 3.5, would imply 3 halvings of $CO_2$, with a correspondingly lower magnitude of change in temperature.



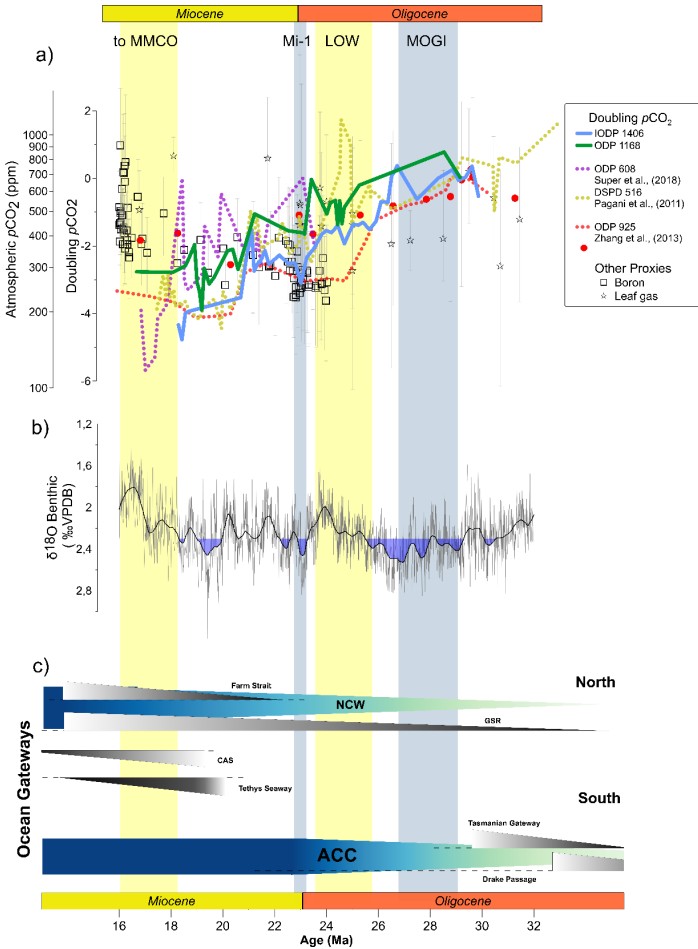

**Figure 8: Implications of CO₂ as main climate driver. a) Shows *p*CO₂ doubling compared to previous estimates compiled in CenCO2PIP Consortium, (2023) (Erdei et al., 2012; Greenop et al., 2019; Liang et al., 2022a; Liang et al., 2022b; Londoño et al., 2018; Moraweck et al., 2019; Reichgelt et al., 2020; Roth-Nebelsick et al., 2014; Sosdian et al., 2018; Steinthorsdottir et al., 2021; Sun et al., 2017; Tesfamichael et al., 2017; Zhang et al., 2013). Only long term records from alkenone derived CO₂ are recalculated.**
**b) Benthic δ¹⁸O global compilation (Westerhold et al., 2020). c) Schematic representation of main paleoceanographic and paleogeographic changes over the studied time interval for the Northern and Southern Hemisphere. ACC: Antarctic Circumpolar Current. NCW: Northern Component Water.**

The late Oligocene climate and CO₂ paradox has been discussed based on previously published lower resolution $\varepsilon_p$ record from Site 925 (O'Brien et al., 2020). Our new results from two additional sites confirm the steep CO₂ decline through the late Oligocene warming and underscore the paradox. On a global scale, biomarker SST estimates do not show evidence for systematic cooling during the CO₂ decline (Guitián et al., 2019; Liu et al., 2018; O'Brien et al., 2020). If the interpretation of $\varepsilon_p$ as a CO₂ decline is correct, it suggests a very different set of feedbacks and climate sensitivity during this time, or widespread regional heat transport effects on regional temperatures or significant misinterpretation of measured biomarker temperature signals. During this time the inferred CO₂ decline also coincides with sequence stratigraphic evidence for ice margin retreat in Antarctica (Levy et al., 2019; Salabarnada et al., 2018), and sea level transgressions inferred from estimates of deep sea δ¹⁸O$_{sw}$ and Mg/Ca records (Miller et al., 2020), suggesting a substantially different relationship between ice expansion and CO₂ than characterized the late Neogene.



For the late Oligocene to early Miocene, the Southern Ocean Site 1168 is the only surface ocean temperature record which exhibits a long-term decline in SST coincident with the record of a large magnitude $pCO_2$ decline and decline in radiative forcing from the greenhouse effect. This long term cooling is despite the equatorward drift of the site over this time interval (Guitián and Stoll, 2021). Potentially, the ODP Site 1168 temperature trend reflects global temperature during the $CO_2$ decline, and whereas the long term alkenone temperature record at Newfoundland Ridge Site 1406 and Site 1404 (Liu et al. 2018) is

affected by variations in the heat transport from the Gulf Stream that overwhelms the signal of radiative greenhouse forcing. While there is no evidence of a concomitant long term cooling in the benthic $\delta^{18}O$ series (e.g. Westerhold et al. (2020)), the $\varepsilon_p$ minimum at 19 Ma coincides with a local maximum in benthic $\delta^{18}O$.

A decoupling was at one time proposed for the late Miocene based on apparent negligible $pCO_2$ change and substantial cooling of SST (LaRiviere et al., 2012). Revisions of the alkenone carbon fractionation to $CO_2$ calibration approaches for low $pCO_2$

periods have refined the record from the last 15 Ma, revealing clear $pCO_2$-SST covariation (Rae et al., 2021; Stoll et al., 2019). However, the Oligocene paradox is not easily resolvable from updated calibration of the $\varepsilon_p$-$CO_2$ relationship because the late Oligocene divergence arises from an inverse correlation between $\varepsilon_p$ and SST reconstructions. Therefore, continued re-evaluation of SST records and interrogation of biogeochemical cycles potentially affecting the growth and physiology of alkenone producers, are needed to reconcile climate sensitivity to $CO_2$ in the Oligocene to early Miocene.

**5 Conclusions**

The new long term alkenone $\varepsilon_p$ records from the Oligocene to early Miocene at North Atlantic Site IODP 1406 and Southern Ocean Site ODP 1168 reveal a significant 8 to 10 ‰ shift. The new records resolve abrupt 3 ‰ declines from 27 to 24.5 Ma and 24 to 22.5 Ma. The long term trend is comparable with previous lower resolution analysis when they are recalculated with the same methodology.

In addition to $CO_2$, $\varepsilon_p$ may be modified by changes in cellular surface area to volume ratio and growth rate regulated by light, temperature and nutrients. However, our assessment of these effects using records of coccolith size and alkenone temperature estimates for exact time intervals of $\varepsilon_p$ determination, shows that size and temperature effects have a negligible impact in the long term declining trend. The similarity of $\varepsilon_p$ in widely separated sites experiencing contrasting temperature histories strongly suggests a global $CO_2$ decline as the most likely cause of the declining $\varepsilon_p$. At the same time, our high-resolution sampling

reveals significant orbital scale variability in $\varepsilon_p$ and underscores the potential for aliasing in low resolution records. Higher resolution $\varepsilon_p$ time series, and more precise age models on legacy $\varepsilon_p$ records to facilitate more confident comparisons of trends among sites, will provide a better characterization of the key long term trends.

Our results highlight the paradox of complex relationships between $CO_2$ indicators and SST at both the orbital and multi-million year timescales. The higher resolution sampling between 28.7 to 29.7 Ma shows that orbital $\varepsilon_p$ maxima do not coincide

with orbital minima in ice volume and/or warmer deep ocean temperature. Similarly, through the late Oligocene warming, $CO_2$ decline contrasts with evidence for Antarctic ice retreat and evidence of stable or warming SST. The transition from late Oligocene to early Miocene, reaching minimum $CO_2$ around 19 Ma, is coincident with significant cooling only in the Southern Ocean Site 1168, but not the North Atlantic site which may be more affected by changes in ocean heat transport.

**Data availability**

Data presented in this paper is stored at Zenodo public repository (https://doi.org/10.5281/zenodo.13908062)



**Author contribution**

Study was conceived by HMS and PJP. Analysis completed by JG, SRP, RSW and LA. Interpretation by JG and HMS. Writing of original draft by JG and HMS with support of PJP.

**Competing interests**

The contact author has declared that none of the authors has any competing interests.

**Acknowledgments**

This paper presents data on sediment samples provided by the Ocean Drilling Program (ODP, IODP). We thank Maddie Santos for lab assistance with biogenic Si determinations. We thank Madalina Jaggi for assistance with carbonate stable isotope
measurements.

**Financial support**

This research was funded by the Swiss National Science FoundationAward 200021_182070 to HMS.

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
