# Peer review of "Oligocene-early Miocene paradox of $pCO_2$ inferred from alkenone carbon isotopic fractionation and sea surface temperature trends"

_Climate of the Past, 2024_

## Author Comment (AC1)

**Comment by Peter Bijl**

I read the manuscript with great interest, it represents an interesting set of new data spanning the Oligocene-Miocene boundary. I am in general happy with the manuscript, but do feel that a lot of published information on Site 1168 is omitted, while these actually shed light on the interpretation of the data. First of all, the dinoflagellate cyst-based oceanographic reconstructions (Hoem et al., 2021; Hou et al., 2023b) shed light on the prevailing ocean conditions at the site: potential changes in upwelling, the ocean zone overlying the site, proximity of the subtropical front, presence of Leeuwin Current Influence, etc. all of which are crucial to at least qualitatively assess growth conditions and thus the ep and pCO2 reconstructions from this site. The papers referenced above would show (by the rather constant dinoflagellate cyst assemblage composition) little latitudinal migration of fronts and the high abundance of Spiniferites evidence for a persistent influence of the Leeuwin Current at the site. Secondly, the high-resolution TEX86-based SST data is also available for Site 1168 (Hou et al., 2023a). Although it makes sense to infer SST from alkenones to infer CO2 and ep, for arguments made in the paper, I think the TEX86 record still has value in the presentation of the oceanography at the site. I understand that this work was the result of a PhD project and that this part of the thesis was finished before the publication of these papers, and in that light it is understandable that the said studies were omitted. I suggest the authors do incorporate this information in a more comprehensive picture of the oceanographic development at ODP Site 1168 so that the whole study becomes more complete.

Also, the ACC development illustration in Figure 8 is somewhat outdated by recent insights that suggest that the modern-strength ACC did not start until the late Miocene (Evangelinos et al., 2022; 2024). Before that time, the ACC remained arguably weak (see, e.g., Sauermilch et al., 2021 for a recent modelling study).

Regards, Peter Bijl

We sincerely appreciate the constructive comment to further discuss the relevance of other data available at Site 1168.

We propose to incorporate in Section 2 Sites and Sediments, the additional background on the oceanographic conditions at Site 1168 as reflected by the dinoflagellate assemblage work reported in Hou *et al.*, 2023.

*"…in contrast, ODP 1168 moved from 55°S to 48°S between 30 and 15 Ma (Torsvik et al., 2012; van Hinsbergen et al., 2015).* **Paleoecological reconstructions from dinoflagellates confirm that the waters above Site 1168 were continually influenced by the Leeuwin Current and located well equatorward of the Subtropical Front (Hou et al., 2023).** *"*

We agree with the reviewer, that the alkenone-derived SST we produced from the same samples as Ep is the most appropriate for calculations and comparison with the Ep and $CO_2$ discussion, because the alkenone SST derives from the same organisms and is matched to the same samples, but that is valuable to mention previously SST trends. Because the SST trends themselves are not a main focus of this paper, we propose now to describe the additional $TEX_{86}$ results (Hou *et al.*, 2022) in section 4.3.1 where we review the long term trends in Ep and SST including:

*"**Unlike alkenone-based SST, published TEX86 SST record at Site 1168 (Hou et al., 2022) does not indicate a transition to lower temperatures from the Oligocene to early Miocene, suggesting different temperature trends in the season or depth niches of the different proxy carriers.** "*

We consider it beyond the scope of this paper to elaborate on potential sources of deviation between the TEX$_{86}$ and Uk proxies and hope that a future publication by GDGTs as well as alkenone experts can further explore this issue.

Additionally, following the comment suggestion we propose to clarify in the new figure and caption that the ACC arrow refers to the shallow (surficial) circulation and is not synonymous with the late Miocene deep ACC described by Evangelinos (2022, 2024).

References used in this comment

Evangelinos, D., et al. (2022). "Absence of a strong, deep-reaching Antarctic Circumpolar Current zonal flow across the Tasmanian gateway during the Oligocene to early Miocene." Global and Planetary Change **208**.

Evangelinos, D., et al. (2024). "Late Miocene onset of the modern Antarctic Circumpolar Current." Nature Geoscience **17**(2): 165-170.

Hoem, F. S., et al. (2021). "Late Eocene-early Miocene evolution of the southern Australian subtropical front: a marine palynological approach." Journal of Micropalaeontology **40**(2): 175-193.

Hou, S., et al. (2022). "Lipid biomarker-based sea (sub)surface temperature record offshore Tasmania over the last 23 million years." Clim. Past Discuss. **2022**: 1-33.

Hou, S., et al. (2023). "Equatorward subtropical front migration and strong dee-sea cooling in the Neogene." Nature Communications **14**: 7230.

Sauermilch, I., et al. (2021). "Gateway-driven weakening of ocean gyres leads to Southern Ocean cooling." Nature Communications **12**(1).

---

## Author Comment (AC2)

Guitian et al., 2024 fills a gap in our understanding of Cenozoic CO2 concentrations, and highlights that the long-term decline in Ɛp in the mid-Cenozoic previously described is global in nature. The study aims to identify global shifts in Ɛp, that are likely to be related to CO2 concentration, by using sites from several contrasting locations to produce a combined Ɛp record. However, this goal is constrained by limitations in the sites' respective age models, which make detailed comparison of the timings of Ɛp changes challenging. The study is well written and illustrates its point well, but would benefit from further integrating its data with existing published records, and with a greater acknowledgement of the limitations of modern calibrations to geologic data. The study does not reconstruct CO2 directly, but reconstructs Ɛp and applies transformations derived from culture studies to account for temperature effects, generating a semi-quantitative CO2 reconstruction. This approach is valid, given the current literature, but a more robust comparison of the semi-quantitative reconstructions with existing δ11B CO2 reconstructions would be of great relevance to its conclusions and place its results better in context. As-is, it is difficult to say on reading this paper whether it agrees or disagrees with existing CO2 reconstructions produced using a technique generally considered robust. The proposed large drop in CO2 through the study interval has profound climate implications if true, as described in the text, but seems at odds with δ11B data, to the best of my knowledge. I would like to see greater integration with existing data from alternative archives and, as the claims of a four-fold CO2 drop and large-scale decoupling of CO2 from temperature over the interval are, as the text admits, paradoxical. It seems as likely to me that the modern culture studies from which the temperature deconvolution is derived are not directly applicable to Oligocene/Miocene alkenone producers, that some other change in algal biology occurred, or that nutrient dynamics shifted more profoundly than biogenic silica records suggest.

We sincerely appreciate the suggestions from the Anonymous Referee #1. As detailed below we propose to incorporate some additional discussion of the existing published records which were plotted in Figure 8. Our manuscript is indeed seeking to call attention to the paradox and we have extended the final paragraph to further emphasize the additional steps that will be needed across the full set of $CO_2$ and climate proxies to reconcile these apparent differences.

Specific points:

• Line 48: there are a lot of boron estimates for the younger half of the interval – I don't think it's accurate to say that most of the existing estimates are marine phytoplankton-derived.

We propose to revise the paragraph of the introduction to clarify:

*The long term $pCO_2$ trends from the Oligocene to early Miocene are derived from the sensitivity of marine algae to $pCO_2$ , while published $δ^{11}B$ based $CO_2$ estimates cover the latest Oligocene into early Miocene (younger than 24 Ma) (Rae et al., 2021).*

• Line 201 - I think you need to discuss the difference in results from using benthic and bulk δ13C for your carbonate measurements. Since your bulk and benthic-derived Ɛps are reconstructed with 2 different values that are ~2‰ apart in Figure 2, have these both been converted to δ13CDIC? That's a large offset. The fact that the monte carlo simulations of the bulk- and benthic- derived Ɛp don't overlap in figure 2 or 4 suggests the uncertainty is much higher than it's been calculated as.

In the methods section (line 205) we described that, similar to previous studies, we adjusted the benthic $\delta^{13}C$ by adding 2 permil to correct for the surface-deep $\delta^{13}C$ gradient and estimate the surface ocean $\delta^{13}C$.

We propose to rearrange this paragraph to clarify.

*We calculate the $\delta^{13}C$ DIC from the $\delta^{13}C$ of the bulk carbonate, which is dominated by Reticulofenestra coccoliths (Guitián et al., 2020). Because there is no divergence of vital effects between small and large coccoliths in the late Oligocene to early Miocene (Bolton and Stoll, 2013), we propose that the offset between coccolith $\delta^{13}C$ and DIC is likely to remain constant. We subtract 0.5 ‰ from the $\delta^{13}C$ bulk to calculate $\delta^{13}C$ DIC, based on average alkenone-producing coccoliths cultured at DIC <4 mM compiled in Stoll et al. (2019). Support for estimating photosynthetic fractionation from coccolith $\delta^{13}C$ is provided by recent culture studies of G. oceanica (Torres Romero et al., 2024). In previous studies, the $\delta^{13}C$ DIC has also been estimated from the $\delta^{13}C$ of calcium carbonate of benthic foraminifera with the assumption of a constant and known offset between the $\delta^{13}C$ DIC of the deep and surface ocean. Site 1406 and 925 features sufficient well preserved benthic foraminifera, mainly epifaunal Cibicidoides spp. larger than 200 μm. For an additional sensitivity test to evaluate the significance of the method of DIC estimation and facilitate comparison to other published Ep records calculated from benthic $\delta^{13}C$, we also estimate surface ocean DIC by adding a constant offset of +2 ‰ to the $\delta^{13}C$benthic measurements, following previous Miocene and Oligocene studies (Guitián et al., 2019; Pagani et al., 2011; Zhang et al., 2013).*

• A single rapid 3‰ Ɛp drop is stated to occur at multiple time intervals in the text – 26.5-25.4 on line 246 (the only drop in that interval is site 1406), 26-24.5 on line 281 (occurs at 925 and possibly at 1168, but not 1406), and then at 27-24.5 in the conclusion. With the age model uncertainty and low resolution in several records, it seems difficult to say whether these were a synchronous event or if the timing differed between sites. At site 516, it looks like Ɛp increases rapidly in (or close to, depending on how it's defined) the same interval, and decreases later. This makes it seem less likely to me that it was a single event – the text states that it's still possible within age model uncertainty, but that's a lot of age models that don't match up well. Are there any potential explanations for a staggered Ɛp drop over several millions of years? Either way, I don't think it's entirely accurate to say it's "resolved" (as in the conclusion line 467).

We propose to revise the text to emphasize the uncertainties imposed by: sedimentation gaps in some sites making it impossible to test the reproducibility of some events in the new records presented here, and the uncertainty in age models of sites which have not been synchronized to a Sr isotope stratigraphy. We also propose to revise the text to refer to the broad interval over which the most abrupt change is observed.

Specifically, we propose to edit lines 281, 467, and 246 to describe the core time interval of this transition with some uncertainties, as 27 to 24.5, which is similar in sites 925 and 1406. The new version of the text will highlight that the lack of Ep measurements prior to 25 Ma in 1168 makes it difficult to evaluate if the transition also occurs in this site.

In 516, we had noted in line 284 " *The late Oligocene at DSDP 516 features a 5‰ peak in Ep between 24.5 and 24.9 Ma, which is not reflected at 1406, or 1168 sites.*"

Regarding this record we propose to clarify that:

*With current information, we cannot assess if this difference reflects age model uncertainty, potential analytical uncertainty from GC-IRMS chromatography, or aliasing.*

• Figure 4 – what happened to site 925? It's on figure 2 but not here.

Following the reviewer suggestion we propose to add 925 to this figure in the new version.

• Figure 6 – that's a lot of correlations that all show a lot of different things, and in many cases have very few data points associated. The time bins are inconsistent lengths of time apart, and contain inconsistent numbers of data points. Several bins contain too few data points for a Pearson's correlation to accurately and precisely constrain trends. Consider excluding bins below a certain number data points, and looking at Cook's Distance, DFITS, or some other measure of influential observations – e.g. the (slightly) positive correlation in Panel (a) at 16-19 Ma probably wouldn't be present if it wasn't for the youngest point. If you're going to show correlation lines then you should show the points they refer to, meaning the temperature-detrended points and temperature/size-detrended points should also be shown, rather than just their trendlines. I'm sure this makes the plot much messier and may require a separate or much larger figure to show properly, and part of it could potentially be moved to the supplementary, depending on what it shows. The dataset relationships shown in Table S3 illustrate the point better, if less interestingly than a nice figure, though they need a description of what the colours mean. I'm also not entirely sure what the grey lines for the Torres et a. (2024) relationship between temperature and Ɛp mean here. All the Ɛp values recorded seem a long way away from this relationship – is this because $CO_2$ was higher in the geologic data than in the culture experiments? Clearly these samples are behaving very differently from the cultures, which makes me wonder if any temperature response data taken from them is applicable. I suspect I'm misinterpreting this and the intended interpretation is more complex, but this needs clarifying in the text.

We appreciate the reviewer's suggestion to simplify this Figure 6. We propose to retain the colors to highlight the individual time windows, but we will remove the individual linear fits from the figure. Then Table S3 will remain as a reference to assess the relationships within each time interval.

Since we feel it is important to show the measured data distribution as well as the trends when the temperature-growth rate effect on Ep is accounted for, we propose to add a separate panel for each site plotting the points with the growth rate-corrected Ep. We agree that this will more transparently illustrate the key points.

With this illustration, it would no longer be required to include the Torres *et al.,* (2024) slope in the figures (in the current version this relationship is deliberately offset from the other data to illustrate the slope).

• Similarly with figure 7 – this one is better, but I'd still prefer it if there was a point representing the temperature-removed reconstructed values, rather than just a line that looks like an error bar at a glance. Is the R2 for the temperature-corrected points similar to that for the raw values?

We appreciate the suggestion of the reviewer and would propose to present Figure 7 then with two sets of panels, the upper set of three with the measured Ep and a lower set of three with the growth-rate corrected Ep (rather than the lines which could be confused with error bars). In this way there will be separate R2 values and their origin will be clear.

• Figure 8: I'm not quite sure how the compound axes with the CO2 doubling work – either there's a direct conversion that can be made to CO2, in which case the doubling CO2 axes aren't needed, or there isn't, in which case the axes shouldn't be compared on the same y axis.

We propose to describe in the figure legend that the pCO2 refers to the boron and leaf gas proxies (in fact, red dots from 925 published $CO_2$ will be omitted). Additionally we suggest clarifying this visually by moving the atmospheric $CO_2$ axis to the right hand side of the graph and could be vertically displaced.

• Please can you add in a paragraph about how your record compares to the boron (and leaf gas exchange) in Figure 8? With the two records overlaid, it's difficult to see if they match well. On close examination, they don't seem to, which is an important finding.

Following the suggestion of the reviewer to further describe the new records relative to the other proxies, first of all, we propose to modify Figure 8 to better reflect proxy records from the same time interval - specifically truncate the included records at 16.1 Ma (the youngest alkenone point) so that it is easier to compare the boron and leaf gas proxies with the alkenones in the period when they overlap and provide more visual clarity. We also propose to add small symbols to the new 1168 and 1406 Ep records to accurately reflect their resolution and to remove the connecting line when there are hiatuses >1 m.y.

Then, as suggested, in section 4.4. after line 429 we propose to add the following paragraph:

*"Similar to phytoplankton proxy records, the available low resolution leaf gas $CO_2$ records suggest a decline in $CO_2$ from the mid to latest Oligocene. However, in contrast to phytoplankton proxy records for a significant long term decline in $CO_2$ from the early Oligocene through mid-Miocene, leaf gas $CO_2$ proxies suggest higher $CO_2$ in the early Miocene than the Oligocene due to a positive shift across the OMT. Boron isotope-based $CO_2$ records from 24 to 18 Ma show significant variability with no clear trend, although the higher density of data around the OMT suggests a $CO_2$ rise from 23 to 20 Ma which may be consistent with the trend observed in the Ep record at Site 1406, which has the highest resolution for this time interval."*

*Additionally, we propose to clarify last section of the discussion in line 461:*

*"However, the Oligocene paradox is not easily resolvable from updated calibration of the εp-CO2 relationship. The late Oligocene paradox arises from an inverse correlation between $\varepsilon p$ and SST reconstructions in regions other than the Southern Ocean such as the North Atlantic, and a lack of correlation between $\varepsilon p$ and the global climate signal in benthic $\delta^{18}O$ trends. The discrepancies between alkenone and published TEX86 at ODP 1168 suggests continued reevaluation of SST proxy interpretation are needed, along with evaluation of the potential influence of changing surface ocean circulation on SST in some locations such as the North Atlantic. Additionally, the divergence of $CO_2$ trends among $\varepsilon p$ and boron isotopes suggest that further interrogation of ocean chemistry biogeochemical cycles potentially affecting the growth and physiology of alkenone producers and the calculation of $CO_2$ from boron isotopes, are crucial to reconcile climate sensitivity to $CO_2$ in the Oligocene to early Miocene."*

Minor points and typos:

• Line 63: 1406 is a bit far north to be subtropical – it's referred to as midlatitude later in the same paragraph as well.

Corrected as suggested in the new version of the text

• There are a few places where references are in the wrong brackets – i.e. (Guitian et al., 2024), rather than Guitian et al. (2024) where they're referred to in the text. I spotted ones on lines 94 and 96, and 226, but there might be more.

We will correct this wrong referencing in the new version of the text

• At line 174 it says that samples 23.1-29./1 Ma in 1168 could not have UK'37 resolved (and therefore, presumably not good enough resolution for alkenone δ13C measurement), but in figure 2, the 1168 record goes up to around 26 Ma – are the dates right?

Most of the samples from 23.1 to 29.1 had not well resolved C37, however some of them this interference was found in the C37.3, and therefore 37.2 was resolved well enough for a δ13C measurement. Detailed description can be found the supplementary table

We propose to clarify the text:

*"The RTX-200 column provided substantially improved resolution of C38 peaks, allowing quantification of C38:2 and C38:3 ME peaks, but for samples between the ages of 23.1 and 29.1 Ma in ODP 1168 it did not perform well enough for all the C37 peaks. Therefore, for this set we provide temperatures estimated from the $U^{k'}_{38ME}$ ratio applying the Novak et al. (2022) core top calibration (Table S1)."*

• Line 184: grammar – "GC oven was set at 90C…"

Will be corrected in the text.

• Figure 4 – the axis lines are sometimes missing, and the tick marks aren't placed right on the age scale at the bottom.

We thank the reviewer for noticing these errors. Are due to the low resolution of the graph file placed within the word text. Will be corrected in the new version.

• Line 380 – GDGTs plural.

Will be corrected in the text.

• Line 419-411 – R2 should be superscripted

Will be corrected in the text.

---

## Author Comment (AC3)

**Referee2**

Guitián et al. built two new records of $\varepsilon_p$ from the Oligocene to early Miocene based on IODP Site 1406 and ODP Site 1168, along with the new $\varepsilon_p$ record from Site 925 to supplement previous published low-resolution record. As $\varepsilon_p$, the carbon isotopic fractionation during the photosynthesis of phytoplankton, is determined by both aqueous $CO_2$ levels and physiological parameters of phytoplankton. To extract variations in atmospheric $CO_2$ levels based on changes in $\varepsilon_p$ from the Oligocene to the early Miocene, the authors evaluated the influence of varying physiological parameters, including cell size and growth rate, on $\varepsilon_p$ evolution. The influence of changing cell size was assessed using measured coccolith sizes and a statistical multilinear regression model developed by Stoll et al. (2019), which shows that $\varepsilon_p$ is a function of aqueous $CO_2$ concentration, light, growth rate, and cell radius. For growth rate, they assumed it is controlled by temperature and used the sensitivity of $\varepsilon_p$ to temperature, as derived from the culture experiments by Torres Romero et al. (2024), to represent the sensitivity of $\varepsilon_p$ to growth rate. They used changes in biogenic silica (bioSi) concentrations in sediments to indicate variations in nutrient concentrations; higher bioSi concentrations suggest increased nutrients and growth rates. They conclude that size and temperature effects have a negligible impact on the long-term declining trend of $\varepsilon_p$, and that a global $CO_2$ decline is the most likely cause of the decrease in $\varepsilon_p$.

Overall, the manuscript reads well, but a lot of phrases/sentences are confusing and difficult to understand. Please look at the specific comments below. In addition, I see several major shortcomings of the analysis and presentation of the results.

We would like to thank the detailed revision of the manuscript provided by the reviewer that we believe substantially improves the discussion of the dataset. We provide a response to the comments below.

First, I do not think it is appropriate to use a single value for temperature-$\varepsilon_p$ sensitivity (0.48‰ decrease in $\varepsilon_p$ per 1°C warming) to represent the sensitivity during the Oligocene to early Miocene. The 0.48‰ decrease in $\varepsilon_p$ per 1°C warming is based on the linear regression of temperature and $\varepsilon_p$ data (22 samples) from culture studies by Torres Romero et al. (2024), conducted at 22 different combinations of temperature, $CO_2(aq)$, and light. Torres Romero et al. (2024) demonstrates that the sensitivity of $\varepsilon_p$ to temperature varies significantly across different $CO_2(aq)$ ranges: 0.37‰ decrease in $\varepsilon_p$ per 1°C warming when $CO_2(aq)$ ranges from 4 to 22.5µmol/kg, and 0.95‰ decrease in $\varepsilon_p$ per 1°C warming when $CO_2(aq)$ ranges from 22.5 to 41 µmol/kg. Given that $CO_2(aq)$ likely fluctuated across these ranges during the Oligocene to early Miocene, the temperature-$\varepsilon_p$ sensitivity may also have varied. Therefore, relying on a single value derived from the culture experiment is not reliable for representing the entire period.

Second, the sensitivity of $\varepsilon_p$ to temperature derived from culture experiments does not necessarily represent the sensitivity of $\varepsilon_p$ to growth rate in real geological environments. Growth rate is influenced by a combination of factors, including light, temperature, nutrient availability, $CO_2$ levels, cell size, and other variables, all of which varied significantly over the geologic past. Although Torres Romero et al. (2024) demonstrates that the temperature-sensitive $\varepsilon_p$ variation can be fully explained by the temperature sensitivity of growth rate, this does not imply that the relationship is directly applicable to real-world conditions of the geologic past.

We appreciate these suggestions from the reviewer. In this study we provided the temperature-$\varepsilon_p$ growth rate correction as a sensitivity analysis and chose the 0.48‰ dependence because this was also shown to be consistent with the growth rate effect on $\varepsilon_p$ in culture studies compiled in Stoll et al., (2019) given the

temperature-growth dependence observed for phytoplankton in the modern ocean (Fielding, 2013). This comparison discussed in Torres-Romero et al., (2024) suggests consistency between field and culture experiments. We acknowledge the diverse slopes found in the culture study and that it is also possible that the modern temperature growth dependence observed by Fielding, (2013) may be different in an ocean with higher $CO_{2[aq]}$.

In this direction and to clarify the interpretation of the dataset in the revised version of the manuscript, we propose to adjust the Figures 6, and 7 in the main text as described in our response to Reviewer 1 (providing a set of panels with measured Ep and a separate set with Ep corrected for temperature). The set with Ep corrected for temperature, we propose to add an error bar illustrating the range in Ep when the slope of the correction ranged from 0.37 to 0.95 ‰ per 1°C warming.

Third, more evidence is needed to justify using changes in biogenic silica (bioSi) content in sediments to represent variations in surface ocean nutrient concentrations, particularly nitrate and phosphate, which are critical for coccolithophore growth. Sedimentary bioSi is primarily linked to the Si biogeochemical cycle, which likely differs from N and P cycles. Additionally, sedimentary bioSi is influenced by factors such as dissolution and preservation, limiting its reliability as a proxy for ocean nutrient concentrations, especially for nitrate and phosphate. Furthermore, the interpretation of bioSi results is inconsistent, with the authors at times referring to bioSi content as the delivery rate (Line 305) and at other times as the burial rate (Line 316).

We fully agree with the reviewer that biogenic silica is an imperfect indicator for surface ocean nutrient concentrations in the sites in the Oligocene. We had initiated the discussion in line 305 with cautious tone " *as one possible nutrient indicator*". Yet, in the absence of any superior indicators of surface ocean nutrient content, we suggest it is worth including biogenic Si in the comparison because nutrient content is one factor which affects the production and export of biogenic Si. We do not feel it is the place to go into discussion about opal export vs burial because opal accumulation rate is widely used in many settings for the Pleistocene as an indicator of opal export because the sediment dissolution is buffered when accumulation rates are high, and this is not the key relevant caveat for the current discussion. Line 305 refers to the delivery as part of the explanation of the proxy process, whereas line 316 refers to the observed burial.

To address this limitation we propose to expand the paragraph beginning in line 305 (new sentences in bold):

*As one possible nutrient indicator, a higher concentration of biogenic silica (bioSi) in sediments may refllect a higher rate of bioSi delivery to the seafloor due to higher export production produced by siliceous organisms (mainly diatoms) in the ocean (Ragueneau et al., 2000). In the modern ocean, regions with abundant dissolved Si in the photic zone are regions also characterized by higher concentrations of macronutrients such as P and N.* **However, bioSi is an imperfect indicator of past surface nutrient content because coccolithophores have a minimal Si requirement, and Si remineralization in the ocean does not occur at the same rate as soft-tissue nutrients such as N and P.** *At IODP 1406, bioSi concentrations generally increase from the Oligocene to earliest Miocene, potentially indicating a gradual increase in the concentration of dissolved Si in surface waters at the site (Fig. 5). If the increase in dissolved silica observed at the North Atlantic is correlated to an increase in dissolved P or N, it could contribute to increase in growth rate, and therefore likely increase in biomass and chlorophyll,which would reduce light in the water column both being part of the observed long term decrease of εp. However, the actual correlation between bioSi and εp is not that strong (Fig. S2), suggesting that while increased nutrient*

*concentrations could contribute to the long-term evolution of εp, the specific steps of εp decline are less likely to be driven by increased nutrients and growth rate.*

Fourth, the linear relationship between $\varepsilon_p$ and SST (or benthic δ¹⁸O) shown for several time slices in Figure 6 is not statistically meaningful, as the sample sizes for most of these time slices are fewer than 10. Therefore, the conclusions drawn from Figure 6 are unreliable.

We fully agree with the reviewer, that many of the time slices do not have significant correlations. Our conclusion drawn from Figure 6 is that across the overall time interval there are weak positive correlations with SST in Site 1168 and weak negative ones with SST in Site 1406.

To further clarify our conclusion, as suggested in response to Reviewer 1, in new Figure 6 we propose to leave the symbols color coded by time interval but illustrate only the single overall correlation in each site (and to provide a separate panel for measured Ep and for growth-rate corrected Ep, each with their single correlation). The correlations by time period are given in Supplementary S3 and we propose to annotate S3 to indicate which relationships have statistical significance. We propose to eliminate the paragraph beginning in line 372 since it will no longer be part of the main text presentation.

Lastly, the relationship between $\varepsilon_p$ and benthic δ¹⁸O at orbital scales (Figures 7b and 7c) does not yield a clear conclusion, as variations in benthic δ¹⁸O are influenced by both deep-water temperature and ice volume. A more meaningful comparison would be between $\varepsilon_p$ and estimated global mean SST (Gaskell et al., 2022; https://doi.org/10.1073/pnas.2111332119) or surface temperature (Evans et al., 2024; https://doi.org/10.1029/2023PA004788).

We thank the reviewer for bringing this discussion to our attention. We would like to mention that Figure 7 presents results on orbitally resolved Ep and δ¹⁸O benthic as well as bulk sediment δ¹⁸O, extracted from the same samples. These are provided to assess the trend in the relationship between Ep and benthic (or surface ocean carbonate) δ¹⁸O, and no absolute comparison of temperature or sensitivity is derived from them in our paper. Consequently, a linear transformation of the benthic δ¹⁸O to global temperature, such as described in previous studies such as Evans et al (2024), would not change the conclusion we make from this figure: data suggest that there is no direct relationship between lower $CO_2$ and colder temperatures. The δ¹⁸O bulk, dominated by surface dwelling coccolithophorids, is not discussed by the references provided. Therefore we propose to leave Figure 7b in the original unit of measured benthic δ¹⁸O.

Specific comments

Comments to the abstract:

Line 12-13: The statement "most based on the phytoplankton carbon isotopic fractionation ($\varepsilon_p$) proxy" is not accurate. Between 25 and 16 Ma, most of CO2 estimates are based on boron isotopes, not alkenone carbon isotopic fractionation.

Line 17: Full name of "Ma" is needed here Line 17-18: "a higher resolution sampling" —higher than what?

Line 20: Please specify "the two sites". Line 20: climate dynamics is a broad concept. Please clarify it.

Line 21-22: This sentence is confusing, especially the phrase 'average earth surface temperature evolution.' Are the authors referring to the global mean surface temperature?

Line 22-23: what does the inverse relationship between $\varepsilon_p$ and benthic δ18O indicate? Line 25: what do "specific time intervals" represent?.

Line 26: this sentence is incomprehensible.

Line 25-27: Confusing. How does the changing cell size and growth rate explain the divergence between $\varepsilon_p$ and benthic $δ^{18}O$?

Line 29: "While $CO_2$ changes likely caused significant changes in radiative forcing" is not connected to the following sentence "SST variation at the examined sites may have been conditioned by regional heat transport".

Line 31: How does "the relationship between benthic δ18O and $\varepsilon_p$" reflect the phasing between ice growth and global temperature?

*We thank the reviewer for carefully revising the abstract details. To address all the specific comments we propose the following text in the new version of the abstract:*

*Atmospheric carbon dioxide decline is hypothesized to drive the progressive cooling over the Cenozoic. However, the long term $CO_2$ record from the early Oligocene to Miocene time interval, derived from the phytoplankton carbon isotopic fractionation (εp) proxy, differs from what is expected to drive the climate observations. Here, we produce two new long-term records of εp over the Oligocene to early Miocene time interval from widely separated locations at IODP Site 1406 and ODP 1168 and increase the resolution of determinations at the equatorial Atlantic ODP 925. These new results confirm a global footprint of εp shift occurring during this interval. Rapid 3 ‰ declines are found from 27 to 24.5 million years ago (Ma) and 24 to 22.5 Ma, and minimum εp is attained at 19 Ma. Between 28.7 and 29.7 Ma at IODP 1406, a 20-30 ky sampling resolution at Site 1406 reveals orbital scale 100 kyr cyclicity in εp. Making use of alkenone-based sea surface temperature (SST) estimates and benthic $δ^{18}O$ estimates extracted from the same samples, we perform a direct comparison with εp to evaluate the relationship with climate. We observe that across the long Oligocene to early Miocene interval, εp is positively correlated to SST only at the southern ocean Site 1168, but not with SST at the North Atlantic Site 1406. Accounting for the temperature-driven growth rate or cell size effects on εp does not lead to stronger correlations between εp and benthic $δ^{18}O$nor stronger correlations between εp and SST at Site 1406. Moreover, at orbital timescale, the relationship between εp and benthic $δ^{18}O$, albeit weak, implies greater ice volume or colder deep ocean at higher $CO_2$. Despite the persistence of climate paradox, the reproducible albeit trends in three widely separated sites, which experienced contrasting temperature evolution and likely experienced different variations in nutrient availability, suggest that a common $CO_2$ forcing is likely the dominant control on the long term trends in εp. Changing ocean heat transport to the North Atlantic may contribute to the observed decoupling of long term Ep and SST in this location.*

Line 34: Please specify "long-term trends". What trend?

*Sentence in the short summary will be clarified: "Records confirm long-term $CO_2$ record but show contrasting relationships with the sea surface temperatures evolution"*

Line 44: please specify what time interval shows "multimillion year warming" and what time intervals shows "cooling trends"

We propose to revise to:

*"However, the long term decline in CO2 estimated by existing proxy records contrasts with the rather stable climatic state with multimillion year warming **(e.g. Late Oligocene Warming)** and cooling **(e.g. Mi1 glaciation)** trends **interpreted** from deep ocean (Cramer et al., 2011; Lear et al., 2000), and surface ocean records (Guitián et al.,2019; Liu et al., 2009; O'Brien et al., 2020) and with estimated Antarctic Ice sheet volume and sea level (Lear et al., 2004;Liebrand et al., 2017; Miller et al., 2020)."*

Line 54: delete "globally"?

We will revise to: "at any given site"

Line 56: References are needed.

We will add here Rau *et al.,* (1996), as well as Stoll *et al.,* (2019), which discusses both growth rate and light.

Line 60: Full name of "m. y." is needed

Will be adjusted in the new version of the text

Line 60: Please specify "two sites on the south American margin".

Will be adjusted in the new version of the text

Line 61: I would add the name of the Site for the additional North Atlantic record

Will be adjusted in the new version of the text

Line 56-63: please reorganize these sentences. The current sentences are not in logic order. I would put the sentence "In this study, we produce a new long-term record of $\varepsilon_p$ over the Oligocene to Miocene time interval at two new, widely separated locations" right after "One approach to evaluate the relative contribution of physiological factors vs CO2 is to produce $\varepsilon_p$ records from sites of widely contrasting oceanographic setting…". The difference of environmental factors (important for physiological factors of coccolithophores) between Site 1406 and Site 1168 should also be clarified in order to make it connected to the previous sentence.

We appreciate the suggestion and propose to revise as a start of a new paragraph and to read:

*"One approach to evaluate the relative contribution of physiological factors vs $CO_2$ is to produce εp records from sites of widely contrasting oceanographic setting, where the $CO_2$ signal may be expected to be common to both locations but the environmental factors such as nutrient availability might not be expected to change in unison. In this study, we produce a new long-term record of εp over the Oligocene to Miocene time interval at two new, widely separated locations: IODP Site1406 in the subtropical North Atlantic off the Newfoundland coast, and ODP 1168 in the Southern Ocean off of Tasmania. We also increase the resolution of determinations at the equatorial Atlantic ODP 925. The existing εp-based $CO_2$ estimations for the Oligocene are derived from ~1 million year resolution measurements from two sites (Site 925 and 516) on the South American margin of the equatorial and South Atlantic; in the early Miocene an additional North Atlantic record (Site 608) provides data (CenCO2PIP Consortium, 2023)."*

Line 68-69: "an indicator of high-latitude temperature and Antarctic ice sheet extent and/or volume" is not accurate. Variations in Benthic δ18O are controlled by changes in both deep-water temperature and ice volume.

We propose to revise to the strict proxy interpretation: "*Variations in benthic $\delta^{18}O$ are controlled by changes in both deep-water temperature and deep ocean $\delta^{18}O_{sw}$ which reflects ice volume.*"

Line 69: what do "These long-term relationships" indicate? "higher resolution" —higher than what?

For clarification we propose to rewrite to:

"*We further measure Ep and benthic $\delta^{18}O$ at approximately 20-30 ky resolution over a series of eccentricity cycles in the early Oligocene at IODP 1406.*"

Line 70: climate dynamics is a broad concept. Please clarify it.

We clarify by proposing to rewrite to "*climate*" to refer the broad climate indicators ($\delta^{18}O$ , SST) used.

Line 74: Please add the full name of CO2[aq].

Full name will be added in the new version of the text.

Line 75: what do "These" refer to?

We will replace it with: "*Physiological factors were initially…*"

Line 80-81: Please add the equation $\varepsilon_p = \varepsilon_f - b/CO2[aq]$, which makes it easy to read.

Following the reviewer suggestion we will add the equation in the new version of the manuscript.

Line 83-87: Some statements are incorrect. Zhang et al. (2013) also applied modern relationships between b and phosphate. Bolton et al. (2016) and Henderiks and Pagani (2007) do not estimate the difference between the modern b value at the site and the paleo-setting b value. Bolton et al. (2016) uses previous formulations of the relationship between cell size and b, which is derived from Henderiks and Pagani (2007).

In this overview of the introduction we stand by the accuracy of the descriptions included. Zhang *et al.*, (2013) applied the range of phosphorus concentrations in the modern surface water above the site to estimate the modern *b* value (from regressions between *b* and phosphate) and applied this modern *b* value to the past calculation of *$CO_2$* . This is equivalent to our concise statement that their study assumed the modern *b*-value for that oceanographic setting remained constant in the past.

The reviewer's assertion that Bolton *et al.*, (2016) use previous formulations of the relationships between cell size and *b* is not a more accurate characterization of the correction applied. Bolton *et al.*, (2016) have generated a curve of variation in *b* using additional productivity indicators, the alkenone accumulation rate and the coccolith Sr/Ca rate (which have not been discussed in Henderiks and Pagani, (2007)) and computed these as variations between the paleo- and modern *b* value. Thus, we believe that our original statement that this work "*estimated the difference between the modern b value at the site and the paleo-setting b value from productivity proxies or proxies for coccolithophore size*" is both concise and accurate for the scope of the introduction section.

In the new version of the text, following the reviewer suggestion we will segregate the reference to Henderiks and Pagani, (2007) to indicate that it exclusively evaluated the relationship between *b* and size

variations (and not other growth rate proxies) to avoid confusion by juxtaposition with the description of Bolton *et al.*, (2016).

We hope that the new revised text address this comment :

*…previous $pCO_2$ calculates have either (1) assumed the modern b-value for that oceanographic setting remained constant in the past (e.g. Zhang et al., 2013), (2) applied modern relationships between b and phosphate and a simulated paleo-surface ocean phosphate concentration at the site (Pagani et al., 2011), (3) estimated the difference between the modern b value at the site and the paleo-setting b value from productivity proxies (Bolton et al., 2016) or (4) applied variation in the b value at the site based on proxies for coccolithophore size (Henderiks and Pagani, 2007).*

Line 87-88: Please add references.

Line 88: The sentence "b term is not well predicted by growth rate, light or cell size alone in a diffusive model" is confusing.

To address both reviewer comments we propose combining with the next sentence:

*"Despite the appeal of this approach, a recent re-evaluation of cultures and field observations suggest the b term is not well predicted by growth rate, light or cell size alone in a diffusive model but that additional effects occur from carbon concentration mechanisms (CCM) on carbon uptake at lower $CO_2$ concentrations, which cause a deviation in the $CO_2$ dependence from the theoretical hyperbolic relationship (Hernández-Almeida et al., 2020; Stoll et al., 2019)."*

Line 93-94: Could the author provide a brief implication of lower Rubisco fractionation?

We suggest rewriting: *"The lower Rubisco fractionation **has implies a lower sensitivity** of εp to CO2 (e.g. as explored in González-Lanchas et al., (2021))".*

Line 96: what does "This approach" refer to? The previous sentence does not mention any approach.Line 96-97: "the observed slope of $\varepsilon_p$ dependence on CO2" is difficult to understand.

We suggest rewriting as: "*A meta-analysis of experimental culture data (Stoll et al., 2019) suggests that εp features a logarithmic dependence on $CO_2$, rather than the hyperbolic dependence implied by (Rau et al., 1997). **This analysis** does not resolve the mechanisms for the form of the observed **relationship between εp and** $CO_2$, but over the range of $CO_{2\ (aq)}$ from 5 to 30 µM, it provides an empirical relationship for interpreting the magnitude of $CO_{2\ (aq)}$ change implied by a given εp change.*"

Line 103: Change "growth rate" to "growth rate $\mu_i$"

Symbol will be added to the sentence as suggested.

Line 107: Please add references after "While cell size can be estimated from coccolith length".

We will add here the Henderiks and Pagani, (2007).

Line 116-117: Please clarify how 0.5 ‰ decrease in $\varepsilon_p$ per 1°C warming is indistinguishable from the prediction of growth rate effect on $\varepsilon_p$? Krumhardt et al. (2017) only demonstrates the increases in sea surface temperature lead to faster coccolithophore growth rates.

Krumhardt *et al.,* (2017) includes a quantification of the temperature effect on growth rates which Torres *et al.*, (2024) show is of the proper magnitude to explain the observed temperature effect on Ep.

To provide a more detailed description we will expand this paragraph including references:

*"Recent culture studies document a 0.5 ‰ decrease in εp per 1°C warming (Torres Romero et al., 2024), and show that this magnitude is identical to the product of εp dependence on growth rate (Stoll et al., 2019) and the modeled temperature dependence of coccolithophore growth rates (Krumhardt et al., 2017) derived from diverse culture and field studies (Fielding et al., 2013; Behrenfeld et al., 2005; Sherman et al., 2016)."*

Line 134: Please specify "higher resolution".

The new text will clarify: *Additionally, 61 samples (at approximately 15 ky sampling) interval for bulk carbonate isotopes were obtained from IODP 1406 within the 29-30 Ma time window, of which 29 were processed for benthic foraminiferal isotopes and 22 yielded biomarkers sufficient for analysis.*

Line 139-140: Replace "The ODP Site 1168 age model" to "The age model of ODP Site 1168". Similar issues occur throughout this manuscript. Please revise accordingly.

This will be revised.

Line 146: what ages do "the two ODP 1168 samples deeper than the Sr isotope measurements" correspond to?

We will detail this refers to two samples deeper than 562 mbsf

Figure 1: what is ODSN?

New caption of the figure will include the complete reference. ODSN refers to Plate Tectonic Reconstruction Service from the Ocean Drilling Stratigraphic Network (https://www.odsn.de/) using the data from Hay et al., (1999).

Line 159, 170, and 189: Change the bold text to normal formatting. Similar issues are present throughout the manuscript. Please revise them accordingly.

This will be adjusted in revision.

Line 171: please specify the age of "the young set of samples"

The new version of the text will detail that this refers to samples younger than 23.1 Ma.

Line 173-176: the sentence flow is not clear. Please reorganize these sentences.

Following the suggestion to improve the organization of this methods description we proposed the revised text:

*"The RTX-200 column provided substantially improved resolution of C38 peaks, allowing quantification of C38:2 and C38:3 ME peaks. For samples between the ages of 23.1 and 29.1 Ma in ODP 1168 the RTX-200 column still did not sufficiently resolve coelutions on the C37:3 peaks.Therefore, for this interval we provide temperatures estimated from the $U^{k'}38ME$ ratio applying the Novak et al. (2022) core top calibration."*

Line 207-208: References are needed.

ODP 1168 evolving water depth is described in the Site and Sediments section referenced in Line 148. Following the suggestion, reference will be included here.

Line 209-210: Guitián et al. (2020) does not demonstrate that the bulk carbonate is dominated by Reticulofenestra coccoliths. Line 212-213: Here the authors assume that $\delta^{13}C$ of bulk carbonates is equivalent to the $\delta^{13}C$ of coccolith. However, they do not provide any evidence to support this assumption.

We thank the reviewer for arising that a clarification is needed here. Guitian *et al.*, (2020) describe that the coccolith fraction is dominated by *Reticulofenestra*. We will additionally clarify that the foraminfera content is very low. For Site 1406 sample content complete description can be found in Guitian *et al.*, (2019). We will revise lines 203-204:

"*Although the foraminifera content in Site 1406 and 925 is very low, features sufficient well preserved benthic foraminifera, mainly epifaunal Cibicidoides spp. larger than…*" and line 206 states: "...*ODP Site 1168 benthic foraminifera were scarce for picking*"

And lines 208-210

"*Consequently, to follow the same approach for all studied records we calculate the $\delta^{13}C$ DIC from the $\delta^{13}C$ measured on the bulk carbonate, which is dominated by calcareous nannofossils, for which previous studies show Reticulofenestra to be the most abundant genera (Guitián et al., 2020)*"

Line 213: The citation should be the original paper, McClelland et al. (2017), rather than Stoll et al. (2019).

The study by Stoll *et al.*, (2019) has aggregated results from cultures of multiple studies including McClelland *et al.,* (2017). Therefore in this case the meta analysis of Stoll *et al.*, (2019) would be the correct citation.

Line 215: Guitián et al. (2019) describes the method for measuring stable isotopes of benthic foraminifera, not bulk carbonate. Before the section "Estimation of aqueous carbon dioxide $\delta^{13}C$", a section describing the method for measuring stable isotopes of bulk carbonate is needed, including details on sample preprocessing.

We thank the reviewer for noticing this gap in the method description. New text will be revised to: **Bulk carbonate and benthic foraminifera** *were measured using analytical techniques described in in Guitián et al., (2019) with the guidelines from Breitenbach and Bernasconi, (2011) for small carbonate samples on a GAS BENCH II Delta V Plus irMS from Thermo Scientific with international (NBS-19 & 18) and in-house carbonate as standards achieving a precision of 0.07 ‰.*

Line 237-238: Do the authors use the value in equation (1) or the linear relationship between $\varepsilon_p$ and cell radius? The slope of the linear relationship between $\varepsilon_p$ and cell radius, derived from a compilation of culture experiments, is certainly different from the value in equation (1). Please specify the sensitivity of $\varepsilon_p$ to cell radius used here and provide justification for its selection.

For this exercise the sensitivity is the referred to the equation (1) following the empirical relationship from the culture dataset assuming only varying size. New text will clarify:

"*We complete a similar exercise for cell radius, calculating the deviation in εp* **only** *relative to the median cell size, for each point using the culture dependence of εp on cell radius shown in* **equation** *(1)*."

Line 247: Replace "26 ma" to "26 Ma".

This will be adjusted in revised text.

Line 249: Replace "from 28.8 to 29.6 Ma" to "from 29.6 to 28.8 Ma". Similar issues occur throughout this manuscript. Please revise accordingly.

These issues will be adjusted in the revised text.

Line 250: "Several ~ 100 ky orbital scale variations of 0.75 ‰ benthic δ18O and bulk δ18O" is incomprehensible.

Will revise to: *"Over several ~ 100 ky orbital cycles, variations of 0.75 ‰ benthic $\delta^{18}O$ and bulk $\delta^{18}O$ are observed, consistent with previous findings of high 100 ky power in benthic $\delta^{18}O$ in other sites during this time period (Liebrand et al., 2017)."*

Figure 2: Did the authors measure δ¹³C of the bulk carbonate for all three sites? The methods section does not clarify which sites were analyzed for bulk carbonate carbon isotopes.

In this study bulk carbonate was measured for all three sites and values are reported in the data supplement. Method section will clarify in line 208:

*"Consequently, to follow the same approach for all studied records we calculate the $\delta^{13}C$ DIC from the $\delta^{13}C$ measured on the bulk carbonate, which is dominated by calcareous nannofossils, for which previous studies show Reticulofenestra to be the most abundant genera (Guitián et al., 2020)"*

Comments to figure 2:

Line 255: Instead of using solid and transparent lines, I recommend using different colors for the lines.

Line 258: White symbols are not visible on this figure; consider using a more visible color.

Lines 259–260: To maintain consistency, I suggest using either 1σ or 2σ for all the error bars.

Figure will be adjusted following reviewer suggestions.

Line 267: Change "an overall low and stable early Miocene" to "overall low and stable values in the early Miocene"?

Text will be revised accordingly.

Line 272: Is Curry et al. (1995) the correct citation? Curry et al. (1995) is the Initial Report for Leg 154, covering ODP Site 925 alone. It does not include DSDP 516 or ODP 608.

We thank the reviewer for noticing the missing references for the age models datums as described in the supplementary material Table S2. Sites 925 and 516 events and source calibrations are detailed in Guitian *et al.*, (2020) and Site 608 are from CenCO2PIP Consortium, (2023). Citations will be included in the revised text.

Line 272: The phrase "As seen in sites 1168 and 1406" is confusing, as this paper does not present the Sr isotopic stratigraphy of Sites 1168 and 1406.

Following the reviewer comment, we revise the text to clarify: "*As previous studies document for sites 1168 and 1406, Sr isotopic stratigraphy can adjust age determinations by 0.5 to 1 Myr. or even up to 2 Myr in a few cases (Stoll et al., 2024).*"

Line 282 and 287: "within age uncertainty of the decrease" and "within the age model uncertainty of the minimum" are difficult to understand.

Following the reviewer comment, we revise the text to clarify:

"*A steep εp decline between at 21 and 20 Ma in Site 516 may be within age uncertainty of the decrease observed **between 20 and 19 Ma** at ODP 1168 and ODP 925*"

"*The characteristic minimum in εp from 18 to 17 Ma is potentially within the age model uncertainty of the 19 Ma minimum in εp at in 1168 and the 18.5 Ma minimum identified at Site 1406.*"

Line 284: "5‰ peak" is confusing. Do the authors mean "5‰ increase"?

Text will clarify this with *" a transient 5‰ positive excursion"*

Line 295: the title is not accurate.

Following the reviewer suggestion, we propose to modify the section name to *" Potential for size and nutrient effects on ep"*

Line 296: The authors have not discussed the effect of CO2 on $\varepsilon_p$ yet.

The background section (section 2) has discussed the influence of $CO_2$ and physiological factors on Ep. Here in the discussion we elect to first estimate the influence of physiological parameters on Ep before assessing the Ep variation which may be due to $CO_2$. We hope that the new section name detailed in the previous comment emphasizes the aim of the section.

Line 296:  replace "cell surface area to volume ratio" with "cell size"

We will revise the text accordingly.

Line 304: Confusing. How does the deeper mixing cause the lower mean light levels?

Deep mixing brings the cells more time into the lower photic zone where light levels are lower. We propose to cite Hernández-Almeida *et al.*, (2020) which discusses this correlation in detail:

Line 316-322: The main point of this paragraph is not clear.Line 318 and 323: Misra and Froelich (2012) do not suggest an increase in erosion and weathering rates from the Oligocene to the early Miocene. In fact, their $\delta^7Li_{SW}$ data show little change from the middle Oligocene to the early Miocene.

We thank the reviewer for suggesting that a clarification is needed in this paragraph. This paragraph outlines the potential interpretations of bioSi burial, and presents the multiple caveats surrounding its interpretation as evidence for an increase in ocean nutrient concentrations. We agree that the data by Misra and Froelcih (2012) show little change, however there is an increase in $\delta^7Li_{SW}$ in the early Miocene and have a data gap from much of the early Oligocene. The sentence refers to the evidence among all three isotopic systems (which have complementary data coverage across the Oligocene to early Miocene) and includes caveats about the interpretation of the isotopic systems (Rugenstein *et al.*, 2019).

To clarify the main point of the paragraph we suggest:

*"The drivers for increasing bioSi burial rates at Site 1406 are not clear. They could reflect a global increase in nutrient delivery or local effects. Important changes in the rate of continental weathering within the Oligocene- early Miocene are often interpreted from the evolution of radiogenic isotopes of Sr, Li and Os (Misra and Froelich, 2012) including the steep rise in $^{87}Sr/^{86}Sr$, although the precise origin of the late Eocene and Miocene increase in $^{87}Sr/^{86}Sr$ remains under discussion (Rugenstein et al., 2019). On a global scale, the nutrient delivery may be conditioned by the riverine supply of P from continental erosion and weathering of P containing minerals. Yet, on the time scales examined in our records, much longer than the residence time of P, the net effect on nutrient concentrations depends on the balance of the supply and the nutrient removal in sediments."*

Line 399: Is "1 ‰ range" typo?

In the specified time interval there is a 1 ‰ range in the $\delta^{18}O$ values as illustrated in Figure 7.

Line 403: what do "these variables" indicate?

We propose to rephrase for clarification: "*… the impact of temperature-stimulated carbon fixation rates is not a significant impact on the relationship between εp* **and SST or $\delta^{18}O$ benthic** *– a temperature-corrected εp record for the 29 to 29.6 Ma interval would still not exhibit an inverse relationship between εp and $\delta^{18}O$ benthic as observed in the late Pleistocene glacial cycles (Hernández-Almeida et al., 2023).*"

Figure 7: The numeric labels on the x- and y-axes (e.g., "2,2" and similar) are difficult to read. Please adjust them to a clearer format, such as "2.2." Similar issues occur in other figures. Please revise accordingly. Figure 7: please add a, b, and c to each panel of this figure.

Labels will be adjusted as suggested.

Line 409: r2=-0.34 is not possible. R-squared is always a positive value.

The typo will be corrected.

Figure 8: Please add the full name of MMCO, Mi-1, LOW, and MOGI.

These will be added to the figure caption.

Line 443: delete very. what do "a different set of feedbacks" mean? Different from what?

Propose to rephrase: *If the interpretation of εp as a $CO_2$ decline is correct, it suggests that climate sensitivity was either significantly weaker so that no appreciable change in global mean surface temperature occurred, or that available paleotemperature records are significantly biased by regional heat transport effects or available paleotemperature estimates reflect a significant misinterpretation of measured biomarker signals.*

Line 446: Please clarify "a substantially different relationship between ice expansion and CO2."

Sentence will be adjusted and word omitted

Line 451-452:  Please delete "and decline in radiative forcing from the greenhouse effect."

Sentence will be adjusted as suggested

Line 453: there is no evidence to support the claim "the ODP Site 1168 temperature trend reflects global temperature". SST change of Site 1168 is likely a regional signal.

We propose rewording to state that we have raised it as a possibility, that among the two temperature time series, 1168 may potentially be more representative of a global trend. With two sites temperature trends, one from sediment drifts in the North Atlantic, there is no a priori reason to assume that the North Atlantic trend is more representative than the 1168 record:

*"If the ODP Site 1168 temperature trend is more representative of global average temperature trends, whereas the long term alkenone temperature record at Newfoundland Ridge Site 1406 and Site 1404 (Liu et al. 2018) is dominated by variations in the heat transport from the Gulf Stream, then the 1168 temperature trend may reflect the signal of radiative greenhouse forcing."*

Line 462: The term "late Oligocene divergence" is not easy to understand. Please consider replacing "divergence" with a clearer term throughout the manuscript to improve clarity.

This section has been reworded as described in the response to Referee #1 including this replacing suggestion.

Line 465: The conclusion section merely repeats the results presented in earlier sections. In addition to summarizing the findings, the conclusion should discuss the broader implications of the results.

We thank the reviewer for the suggestions to include in the conclusions section. However, we have introduced the broader implications of the results and suggestions for next steps in section 4.4. We follow the style suggestions from the EGU Journals Webinar of Ken Carslaw, that the Conclusion should not feature further discussion but summarize the findings.

Figures and supplementary figures: The current color scheme, particularly the use of red and green in the same figure, is not color-blind-friendly. Please adjust the colors to enhance accessibility and readability for all readers.

Following the reviewer some of the colors in the figure symbol will be revised. In Figure 4 the squares will be varied in size so that they are distinguishable by features other than color. Figure 8 will be revised with different dash patterns for the doubling $p\mathrm{CO}_2$ lines. Additionally, figures will be tested through the color blindness simulator again to revise the color scheme accordingly.

References included in this reply not previously included in the manuscript:

Behrenfeld, M. J., Boss, E., Siegel, D. A., and Shea, D. M. (2005). Carbon-based ocean productivity and phytoplankton physiology from space. *Glob. Biogeochem. Cycles* 19. doi:10.1029/2004GB002299

Fielding, S. R. (2013). Emiliania huxleyi specific growth rate dependence on temperature. *Limnol. Oceanogr.* 58, 663–666. doi:10.4319/lo.2013.58.2.0663

Hay, W. W., DeConto, R. M., Wold, C. N., Wilson, K. M., Voigt, S., Schulz, M., ... & Söding, E. (1999). Alternative global Cretaceous paleogeography.

Sherman, E., Moore, J. K., Primeau, F., and Tanouye, D. (2016). Temperature influence on phytoplankton community growth rates. *Glob. Biogeochem. Cycles* 30, 550–559. doi:10.1002/2015GB005272

---

## Author Response (AR2)

Thank you to the authors for the revised manuscript. I appreciate the effort that has gone into the revision, and think the manuscript is much improved as a result. My only comment for this round is that it would be good if the equation for generating the CO2 doublings in figure 8 was written out explicitly, rather than just explained narratively within the text. Alternatively, if a previous paper that has used this approach already has the equation written out, it could be referenced. As-is, it isn't immediately obvious how you go from a dependence of "In(CO2(aq) of 2.66" plus the temperature correction to the axes of the figure expressed in doubling/halving of CO2. Presumably you're normalising to the mean of the dataset to do so?

We have clarified the CO2 doubling calculations. A paragraph and equation 7 has been added at the end of section 3.3 aiming to detail the relationships:

"From the  $\varepsilon$ p time series we estimate the change in CO2 relative to the maximum values at 29 Ma, using the adjustment in  $\varepsilon$ p for temperature sensitive growth rate described in the previous paragraph, and Eq. (3) as applied in (González-Lanchas et al., 2021), where I reflects the size and light influences on  $\varepsilon$ p and is assumed constant across all time intervals, and the  $\varepsilon$ p dependence on  $\varepsilon$ p ln [CO2[aq]] of 2.66 is the 50th percentile estimate of the modern cultures. We then estimate the doubling/halving of CO2 relative to the CO2 at the reference age (R) applying the solubility for the measured temperature (Zeebe and Wolf-Gladrow, 2001) which can be reduced to:

doubling
$$CO_2 = \frac{\varepsilon_p(t) - \varepsilon_p(R)}{2.66 \ln 2} + \log_2 \left(\frac{\operatorname{sol}(R)}{\operatorname{sol}(t)}\right)''$$

Additionally clarified at section 4.4:

"Although the calculation of absolute CO2 concentrations from  $\varepsilon p$  in the Oligocene and early Miocene remains challenging, the logarithmic dependence of  $\varepsilon p$  on CO2[aq] observed in cultures allows us to estimate the relative changes in CO2 if the sensitivity of  $\varepsilon p$  to CO2 in the Oligocene were similar to modern cultured species **using Eq. (7).** If we incorporate a temperature correction and apply the 50th percentile estimate of the modern culture  $\varepsilon p$  dependence on  $\varepsilon p$  in [CO2[aq]] of 2.66, it implies major changes in CO2 concentrations, with potentially 4 halvings of CO2 concentration from 29 to 16 Ma (Fig. 8)"

Figure 8 footnote now specifies the reference date for CO2 doubling of the sites plotted:

"Implications of CO2 as main climate driver. a) pCO2 doubling for the discussed sites from  $\varepsilon p$  referenced at 29 Ma (Site 608 referenced to maximum at 23 Ma). Solid lines are calculated using the SST corrected  $\varepsilon p$ "

Other minor comments and typos: Line 87: b should be italicised

Corrected

Line 217: "Although the foraminifera content in Site 1406 and 925 is very low, features sufficient well preserved benthic foraminifera, mainly epifaunal Cibicidoides spp. larger than 200  $\mu$ m." Sentence needs restructuring?

Text adjusted: "Although the foraminifera content in Site 1406 and 925 is very low, sediments feature sufficient well preserved benthic foraminifera, mainly epifaunal Cibicidoides spp. in the size range larger than 200  $\mu$ m."

Figure 3: x-axis labels should be below the figure.

Corrected

**Anonymous referee #2**

I appreciate the authors' efforts in addressing my previous suggestions. However, some of my minor comments were not clearly addressed in their response. For example, they often state "will be adjusted" without showing the actual modifications. As a reviewer at least in my case—I prefer to see the specific changes directly in the response, rather than having to search through the revised manuscript to verify them. Most of my comments have been adequately resolved. However, one important point remains unaddressed: the relationship between  $\epsilon p$  and benthic  $\delta 180$  at orbital scales (Figures 7b and 7c) does not yield a clear conclusion. Even in the revised manuscript, the abstract states "at orbital timescale, the relationship between  $\epsilon p$  and benthic  $\delta 180$ , albeit weak, implies greater ice volume or colder deep ocean at higher CO2". This statement remains vague, and the authors do not offer a definitive interpretation. As the authors interpret ep variations as the change in atmospheric CO2 levels, comparing  $\epsilon p$  with benthic  $\delta 180$  evolution, a signal of global climate change, is reasonable to evaluate the global impact of CO2 changes. However, without decomposing benthic  $\delta$ 180 signal into ice volume/sea level component and deep ocean temperature component, it is unlikely that a meaningful conclusion can be drawn from this comparison. Instead, comparing sp with the estimated global mean SST (Gaskell et al., 2022; https://doi.org/10.1073/pnas.2111332119) or surface temperature (Evans et al., 2024; https://doi.org/10.1029/2023PA004788) would likely provide more direct insights, as these records more directly reflect the climatic signals the authors aim to assess.

We recognize the value and appeal of conversions of benthic  $\delta$  18O to estimated global mean SST or surface temperature from the approaches of the suggested references.

On the orbital scale, our submitted manuscript describes the trends between  $\epsilon p$  and  $\delta^{18}O$  benthic, not specific climate sensitivities. The interpretation of this observed trend is unchanged if the  $\delta^{18}O$  benthic signal is partitioned into a temperature and ice volume component using one of the proposed slopes in Evans et~al., (2024). Because we have not quantified absolute  $CO_2$  concentrations over these orbital cycles, we believe there is little added value in attempting an uncertain partitioning of the 1406 benthic signal into ice volume and temperature components at this stage. As further constraints improve quantitative  $CO_2$  interpretation from  $\epsilon p$  during this time period, and detailed geochemical studies, such as Brzelinski et~al., 2020 (deconvolving the benthic  $\delta^{18}O$  with benthic Mg/Ca during a younger Oligocene interval at 1406), provide further support for deep temperature and ice volume deconvolutions, more quantitative interpretation of the ice volume and  $CO_2$  relationships should become possible.

Brzelinski, Swaantje, André Bornemann, Diederik Liebrand, Tim E. van Peer, Paul A. Wilson, and Oliver Friedrich. "Large obliquity-paced Antarctic ice-volume fluctuations suggest melting by atmospheric and ocean warming during late Oligocene." *Communications Earth & Environment* 4, no. 1 (2023): 222.

Specific comments

Line 12: please specify "what is expected to drive the climate observation"

Text adjusted.

Line 30: what long-term CO2 trend? Please specify it.

Text adjusted.

Line 32: temperature and nutrients are considered as environmental factors, rather than physiological factors.

Text adjusted.

Line 44: how did estimated Antarctic ice sheet volume and sea level evolve? How did their evolution contrast with the long term decline in CO2?

Text adjusted.

Line 56-57: environmental factors are not consistent with 'physiological factors' mentioned before.

Now reads as: "One approach to evaluate the relative contribution of physiological factors vs CO2 is to produce  $\varepsilon p$  records from sites of widely contrasting oceanographic setting, where the CO2 signal may be expected to be common to both locations but the environmental factors **affecting the fractionation** such as nutrient availability might not be expected to change in unison"

Line 64: full name of m.y. is needed here.

Corrected

Line 83-89: The term is inconsistently written as b value, b value, b-value, and bvalue. Please choose one format and use it consistently throughout the text.

Corrected

Line 153: a period is needed after (Hou et al., 2023b)

Adjusted

Line 240: full name of GDGT is needed

Corrected

Line 242: change 'GDGTS' to 'GDGTs'

Corrected

Line 282: does correlation indicate correlation between the study sites? Please clarify it.

We now clarify: "The inference of rapid declines is also affected by the age models and the correlation of rapid  $\varepsilon p$  shifts among different sites might be hindered by uncertainties in chronology among the different sites"

Line 418-419: Figure 7e should be referenced after "a temperature-corrected  $\epsilon p$  record for the 29.6 to 29 Ma interval would still not exhibit an inverse relationship between  $\epsilon p$  and  $\delta 180$  benthic"

**Corrected**

Line 441: referenced are needed for modern climate sensitivity

Modern climate sensitivity has been adjusted and referenced to IPCC report Chapter7

Line 442: The estimates of 12 to 20°C cooling from Oligocene to early Miocene seems quite large and may be overestimated, especially considering that Early Eocene global mean surface temperature was only about 10-16°C warmer than pre-industrial levels (Inglis et al., 2020).

Adjusted for clarity and accounting for the broader uncertainty estimates of climate sensitivity of 2 to 5°C:

"Modern General Circulation Models (GCM) summarized by the IPCC estimate climate sensitivity as "very likely" in the range of 2 to 5°C per doubling or halving of  $CO_2$  (IPCC AR6 Assessment, Forster et al., 2021), which if representative for the Oligocene to early Miocene, would imply 8 to 20°C of cooling of earth's mean surface temperature (6 to 15°C incorporating the lower confidence interval of modern culture  $\varepsilon_p$  dependence on In  $[CO_{2[aq]}]$  of 3.5, which would imply 3 halvings of  $CO_2$ ). Although ocean is 70% of the globe and temperature changes are around 1.5-fold less than land temperature (Sutton et al., 2007), such a temperature change of at least 6C would be expected to be reflected in paleoceanographic proxies."

Line 476-477: is there any evidence to support that the temperature trend of ODP Site 1168 is more representative of global average temperature trends

We propose to add the following caveat to the end of the paragraph:

"Yet, temperature trends at either site may be subject to both global factors as well as regional temperatures, and with only two sites with temperature records paired to  $\varepsilon_p$  proxy records it is difficult to ascertain which, if any, of the sites may better reflect global temperature forcing."

We note that the subsequent paragraph already highlighted the need for further temperature records to clarify this effect:

"The discrepancies between alkenone and published TEX86 at ODP 1168 suggests continued reevaluation of SST proxy interpretation are needed, along with evaluation of

the potential influence of changing surface ocean circulation on SST in some locations such as the North Atlantic."

Figures and supplementary figures: in several figures, the y-axis is labeled as "Ep", but it should use the Greek letter epsilon p ( $\epsilon p$ ) to remain consistent with the notation used in the main text. Please update the labels accordingly.

Notation adjusted.

Figure 2 and Figure S1: Using red and green in the same figure is not color-blind-friendly. Please adjust the color scheme.

Symbols were adjusted.